# Ugi Reaction on α-Phosphorated Ketimines for the Synthesis of Tetrasubstituted α-Aminophosphonates and Their Applications as Antiproliferative Agents

**DOI:** 10.3390/molecules26061654

**Published:** 2021-03-16

**Authors:** Adrián López-Francés, Xabier del Corte, Edorta Martínez de Marigorta, Francisco Palacios, Javier Vicario

**Affiliations:** Departamento de Química Orgánica I, Centro de Investigación y Estudios Avanzados “Lucio Lascaray”, Facultad de Farmacia, University of the Basque Country, UPV/EHU Paseo de la Universidad 7, 01006 Vitoria-Gasteiz, Spain; adrian.lopez@ehu.eus (A.L.-F.); xabier.delcorte@ehu.es (X.d.C.); edorta.martinezdemarigorta@ehu.eus (E.M.d.M.)

**Keywords:** multicomponent synthesis, Ugi reaction, α-aminophosphonates, tetrasubstituted carbons, antiproliferative effect

## Abstract

An Ugi three-component reaction using preformed α-phosphorated *N*-tosyl ketimines with different isocyanides in the presence of a carboxylic acid affords tetrasubstituted α-aminophosphonates. Due to the high steric hindrance, the expected acylated amines undergo a spontaneous elimination of the acyl group. The reaction is applicable to α-aryl ketimines bearing a number of substituents and several isocyanides. In addition, the densely substituted α-aminophosphonate substrates showed in vitro cytotoxicity, inhibiting the growth of carcinoma human tumor cell line A549 (carcinomic human alveolar basal epithelial cell).

## 1. Introduction

In the interdisciplinary research field of chemical biology and drug discovery, diversity-oriented synthesis is an interesting model for the production of large chemical libraries of small molecules, bearing multiple functional groups, in order to explore their influence into the biological properties of those substrates [1,2,3]. At the heart of this concept, multicomponent reactions (MCRs) have become a mainstay of medicinal and organic chemistry that allow the preparation of a broad spectrum of compounds with a reduced number of synthetic steps [4,5]. In such synthetic procedures, three reactants or more are combined in the same pot to generate a new substrate, whose structure shows portions of all the starting materials. The atom economy, efficiency, mild conditions and high convergence of MCRs justify a central place in the toolbox of diversity-oriented synthesis [6,7]. Among the numerous MCRs described along the last decades, the Ugi reaction [8] has been verified as one of the most important multicomponent processes for the synthesis of peptide-like structures [9,10]. The Ugi reaction consists on a nucleophilic attack of an isonitrile **6** to an iminium ion **5**, a salt composed of a carboxylic acid **4** and an imine **3**, which is often generated in situ from a carbonyl derivative **1** and an amine **2**. Then, a second nucleophilic attack of the carboxylate anion in the intermediate nitrilium species **7** results in the formation of acyl imidate **8**. The reaction ends with an irreversible Mumm rearrangement of species **8**, leading to α-amido amide substrates **9** in a very efficient manner (Scheme 1). Remarkably, the whole reaction is driven by the Mumm rearrangement since all other species involved in the mechanism are in equilibrium. Due to its versatility, the Ugi reaction has become increasingly practical in the synthesis of many active complex drugs and natural products [11,12,13].

On the other hand, the α-aminophosphonic acid framework enjoys significant attention in medicinal sciences, due to its unique ability to mimic the transition state of peptide cleavage in an irreversible fashion, thus blocking very efficiently enzymes implicated in proteolysis processes (Figure 1). For this reason, α-aminophosphonic acid derivatives and their phosphapeptides display an assorted biological activity, including anticancer properties [14,15,16,17,18]. α-Aminophosphonic acids can be considered as structural isosters of α-aminoacids, where the flat carboxylic acid group has been replaced by a phosphonic acid group, and one of the most straightforward methods for the preparation of both compounds, α-aminoacids and α-aminophosphonic acids, consists on the addition of carbon nucleophiles to α-iminoesters or α-iminophosphonates, respectively [19,20]. While an Ugi reaction using α-iminoesters to afford α,α-diamino acid derivatives is documented [21,22] no examples are described using α-iminophosphonates as starting materials. In addition, the use of ketones or ketimines as substrates in such reactions, in order to generate structures bearing tetrasubstituted carbons, entails additional obstacles, since the inherent steric factors observed in these systems enhance the difficulty level in these synthetic methodologies [23]. In addition, the use of acyclic ketones typically requires preformation of the imine intermediate in a separate step, and the yields of the Ugi are often modest [24,25,26].

In this context, during the course of our research on the addition of nucleophiles to α-ketiminophosphonates, in the past, we achieved the synthesis of tetrasubtituted α-aminophosphonates [27] using cyanide [28], organometallics [29] and nitromethane [30] as nucleophiles and, more recently, we have reported the first enantioselective Reformatsky reaction using acyclic ketimines as substrates [31]. Continuing with our interest in the chemistry of organophosphorus compounds, we thought that α-ketiminophosphonates would be excellent substrates in Ugi reactions for the generation of phosphorated peptide-like structures bearing tetrasubstituted carbons. Due to the great occurrence of tetrasubstituted carbons in natural products and drugs [32], the high affinity of α-aminophosphonates to proteolytic enzymes and the synthetic versatility of multicomponent reactions, a synthetic protocol of an Ugi reaction using α-phosphorated ketimines would be of great value in organic and medicinal chemistry.

## 2. Results and Discussion

### 2.1. Chemistry

*N*-tosyl α-ketiminophosphonates **10** can be synthesized by a formal oxidation of trisubstituted aminophosphonates as reported in literature [28,31]. In our first experiment we studied the Ugi reaction of *N*-tosyl ketimine **10a** (R^1^ = Me, R^2^ = Ph) with phenyl acetic acid **11** and cyclohexyl isocyanide **12a** (R^3^ = Cy) under the typical reaction conditions (Scheme 2). After stirring a mixture of the three compounds in CH_2_Cl_2_ at room temperature for 1 h, NMR showed the complete disappearance of the starting materials and formation of tetrasubstituted α-aminophosphonate **13a**. Due to the insolubility of the starting materials, the use of other environmentally friendly solvents led to the formation of substrate **13a** in lower yields and longer reaction times.

With this result in hand, next we extended the Ugi protocol to different α-iminophosphonates **10** and isocyanides **12** using phenylacetic acid **11** in CH_2_Cl_2_ (Scheme 2). First, different isocyanides **12** were tested in the reaction using ketimine **10a** (R^1^ = Me, R^2^ = Ph) derived from dimethylphosphonate. The reactions proceed fast (1 h) and with good yields, not only using cyclohexyl isocyanide **12a** (R^3^ = Cy), but also with methyl isocyanoacetate **12b** (R^1^ = CH_2_CO_2_Me) or benzyl isocyanide **12c** (R^3^ = Bn) to afford α-aminophosphonates **13b**–**c** (Scheme 2).

Next, diethyl, dibenzyl and di-*iso*-propyl phosphonate substituted ketimines **10b**–**d** (R^1^ = Et, Bn, *^i^*Pr,) were tested as electrophilic substrates with very good results but different reactivity. In the case of diethylphosphonates **13d**–**f** (R^1^ = Et, R^2^ = Ph), and dibenzylphosphonates **13g** (R^1^ = Bn, R^2^ = Ph) the reactions proceed to full conversion after 6 h and even longer reaction times of 14 h are needed for di-*iso*-propylphosphonates **13h**–**j** (R^1^ = *^i^*Pr, R^2^ = Ph) (Scheme 2). These differences in the reactivity related to the size of the phosphonate substituents are in agreement with what has been observed in similar reactions [28,30].

Then, the scope of the reaction was extended to the use of phosphorated ketimines bearing substituted aromatic rings. Aromatic ketimines holding strong electron withdrawing substituents such as a *para*-nitro group showed very good reactivity and aminophosphonate **13k** was obtained in very good yield after 1 h at room temperature (Scheme 2). The reaction is also fast using ketimines with halogenated aromatic groups. Several halogen substituted aromatic ketimines were successfully used in the reaction, including *para*-substituted aromatic rings containing bromine or chlorine to yield halogenated α-aminophosphonates in full conversion after 1 h (Scheme 2, **13l**–**m**). The reaction tolerates also the presence of an *ortho*-fluor substituted aromatic ring in (Scheme 2, **13n**) and even the existence of a perfluorinated phenyl group (Scheme 2, **13o**). Besides, when aromatic ketimines substituted by electron donating groups were used as substrates, an increase in the reaction times was observed. However, α-aminophosphonates **13p**–**q** were obtained in full conversion after 14 h (Scheme 2).

Tetrasubstituted α-aminophosphonates **13** were characterized on the basis of their ^1^H, ^31^P, ^19^F and ^13^C NMR, IR spectra and high-resolution mass spectra (see Appendix A for the detail). For example, ^1^H NMR spectrum α-aminophosphonate **13a** presents the signals corresponding to the aliphatic cyclohexyl moiety with several chemical shifts in the interval δ_H_ = 0.92–1.91 ppm for the five methylene groups and an additional multiplet at δ_H_ = 3.77 ppm for the CH bonded to the nitrogen. The phosphonate moiety is seen as two representative doublets at δ_H_ = 3.80 ppm (^3^*J*_PH_ = 10.5 Hz) and δ_H_ = 3.99 ppm (^3^*J*_PH_ = 10.7 Hz), typical for the diastereotopic methoxy groups at the phosphonate. The presence of the tosyl group is evident from the chemical shift for its *para*-methyl substituent at δ_H_ = 2.33 ppm, that appears as a singlet, and the two doublets at δ_H_ = 7.00 and 7.16 ppm (^3^*J*_HH_ = 8.3 Hz), corresponding to the four aromatic protons, that appear partially overlapped with the five protons of the phenyl substituent in the interval at δ_H_ = 6.99–7.25 ppm. The sulfamide and amide NH protons appear as two doublets that interchange with D_2_O at δ_H_ = 6.47 ppm (^3^*J*_PH_ = 8.2 Hz) and δ_H_ = 6.76 ppm (^3^*J*_HH_ = 6.4 Hz), respectively. Due to the low interchange rate in such acidic protons, the signal corresponding to the NH of the sulfamide moiety is coupled with the magnetically active phosphorus atom, while the amide NH is coupled with the neighboring CH of the cyclohexyl group.

In addition, in the ^13^C NMR spectrum of α-aminophosphonate **13a**, the cyclohexyl group can be detected by the chemical shift at δ_C_ = 49.8 ppm, corresponding to its methyne group, bonded to the nitrogen atom and, due to the stereogenic center present in the structure, the other five methylene carbons show five different signals at δ_C_ = 24.5, 24.6, 25.4, 32.1 and 32.3 ppm. Here, again, the two diastereotopic methoxy groups at the phosphonate moiety are seen as two doublets at δ_H_ = 55.8 ppm (^2^*J*_PC_ = 8.2 Hz) and δ_C_ = 55.2 ppm (^2^*J*_PC_ = 7.5 Hz). The most characteristic chemical shift of α-aminophosphonate **13a** in ^13^C NMR is certainly the doublet corresponding to the quaternary carbon directly bonded to the phosphonate that appears at δ_C_ = 68.5 ppm and presents a strong coupling with the phosphorus atom (^1^*J*_PC_ = 157.2 Hz). The presence of the tosyl group is here deduced from the chemical shift corresponding to its *para*-methyl substituent at δ_C_ = 21.6 ppm and the aromatic carbons with two signals at δ_C_ = 126.5 and 129.1 ppm for each of the two couples of the equivalent CH carbons of the aromatic ring, as well as another two signals for the two quaternary carbons at δ_C_ = 142.4 and 139.2 ppm, the latter seen as a doublet due to the coupling with the phosphorus atom (^4^*J*_PC_ = 1.6 Hz). In the aromatic region it also appears the chemical shifts of the carbons corresponding to the phenyl ring, with the signals corresponding to the two pairs of equivalent CH carbons at δ_C_ = 127.9 and 130.2 ppm, the second as a doublet coupled with the phosphorus atom (^3^*J*_PC_ = 8.3 Hz). The fifth aromatic CH appears at δ_C_ = 128.7 ppm and the quaternary carbon as a doublet at δ_C_ = 131.9 (^2^*J*_PC_ = 1.8 Hz). Surprisingly, the amide carbonyl group does not show coupling with the phosphorus atom and the signal appears as a singlet at δ_C_ = 166.1 ppm.

The most relevant absorptions observed in IR spectrum correspond to the amide, sulfamide and phosphonate moieties. The stretching vibration of amide and sulfamide NH groups can be observed at ν = 3426 and 3333 cm^−1^, respectively. In addition, two strong bands are observed at ν =1678 and 1256 cm^−1^, correspond to the vibration of amide C=O and phosphonate P=O bonds. Finally, the spectrum shows two characteristic absorptions ν = 1333 and 1164 cm^−1^ that correspond to the asymmetric and symmetric stretching vibration of the sulfonyl group.

Regarding the mechanism of the reaction, we theorized that compounds **13** might be formed by a typical three-component Ugi reaction that leads to the formation of the predicted phosphorated α-amido amide **15**, followed by a spontaneous cleavage of the acyl group, due to the high steric hindrance present in the intermediate **15** (Scheme 3). In fact, the same behavior has been observed in the acylcyanation reaction of *N*-tosyl ketimines **10** (R^2^ = Ar, PG = Ts) with pyruvonitrile [28]. In our attempts to detect the acylated intermediate **15**, different carboxylic acids were used in the reaction, but α-aminophosphonate **13a** was obtained in all cases, even when acetic, trifluoroacetic or benzoic acid were used as reagents. Nevertheless, the reaction does not proceed in the absence of a carboxylic acid, which at least indicates that the formation of iminium species is crucial prior to the nucleophilic attack of isocyanide.

In order to check if the Mumm rearrangement was indeed taking place, next we used *N*-trityl aldimine **14** (R^2^ = H, PG = CPh_3_) [33] as the electrophile substrate, in the presence of phenylacetic acid **11** and cyclohexyl isocyanide **12a** (R^3^ = Cy) (Scheme 3). Due to the utilization of an aldimine derived electrophile in the reaction, a less hindered structure is expected in the Ugi adduct, which may result in the isolation of species **15**. However, in this case, trisubstituted α-aminophosphonate **16** was obtained in full conversion, where, the formation of α-amido amide **15** is followed by a spontaneous cleavage of the bulky trityl protecting group (Scheme 3).

Although this last experiment supports an Ugi three-component mechanism of the process, still we were skeptical about the real role of the carboxylic acid in the system. It is true that, considering the accepted mechanism for the Ugi reaction, only through the irreversible Mumm rearrangement all the equilibrium in the process can be displaced to the final products. But yet, it might be vaguely possible that, in the case of our ketimines **10**, a simple addition of isocyanide to iminium species could afford tetrasubstituted α-aminophosphonate **13a** after an irreversible hydrolysis of the nitrilium intermediate, due to the presence of traces of water in the solvent. Then the key question to be addressed is: is the third reactant of the multicomponent reaction a carboxylic acid or is it just water?

This matter could be resolved in view of the fact that the isolation of intermediate **15** was achieved when *para*-fluorophenyl or *para*-trifluoromethylphenyl substituted α-phosphorated ketimines **10l**,**m** (R = CF_3_, F) were used as the electrophile unit in the Ugi reaction. Using phenylacetic acid **11** and cyclohexyl isocyanide **12a**, phosphorated α-amido amides **15a**,**b** were obtained, without the elimination of the amide group (Scheme 4). Although substrate **15b** proved to be very stable, trifluoromethyl substituted α-amido amide **15a** underwent spontaneous hydrolysis of the amide under the air moisture to yield tetrasubstituted α-aminophosphonate **13r**.

NMR properties of phosphorated α-amido amides **15** were very similar to the parent substrates **13** except for some significant differences. In the case of substrate **15b**, the presence of benzylamide group was evident in ^13^C NMR by the existence of the chemical shifts for two carbonyl groups at δ_C_ = 176.4 and 165.3 ppm and a methylene carbon at δ_C_ = 45.7 ppm (DEPT). Key features for this compound in ^1^H NMR spectrum are mainly the two diastereotopic protons of the benzyl group that appear as doublets at δ_C_ = 3.92 and 4.16 ppm with a strong geminal coupling constant ^2^*J*_HH_ = 17.1 Hz. It is also noteworthy the presence of an atypical doublet for two equivalent aromatic protons at δ_C_ = 8.26 ppm (^3^*J*_HH_ = 7.9 Hz) that corresponds either to the benzyl or the tosyl moiety that appears especially deshielded, which is probably originated by the proximity of both aromatic rings due to the steric crowding present in the structure.

In order to shed more light on this issue, we set up an additional experiment where the three-component reaction was performed using of *N*-tosyl ketimine **10a**, thioacetic acid **17** and cyclohexyl isocyanide **12a** in CDCl_3_. However, after 1h at room temperature a complex mixture was observed in the reaction vessel. We hypothesized that the high steric hindrance due to the presence of the tetrasubstituted carbon together with the higher Van der Waals radius of the sulfur atom versus the oxygen (180 pm vs. 152 pm) could be the reason of such different behavior.

For this reason, next we tried the Ugi reaction using a less sterically demanding isocyanide such as methyl isocyanoacetate **12b** (Scheme 5). In this case, formation of thioamide **18** was observed in full conversion. The presence of a sulfur atom in the structure confirms unambiguously the Ugi mechanism of our reaction through the formation of iminium species **19** from α-ketiminophosphonate **10a** and thioacid **17**, followed by a nucleophilic attack of isocyanide **12b**. Then, a second nucleophilic attack of thiocarboxylate anion in the intermediate nitrilium species **20** results in the formation of acyl thioimidate **21**. To complete the Ugi sequence, the acyl transfer from thioimidate **21** to the adjacent nitrogen atom yields irreversibly phosphorated α-amido amide **22** that, due to the high steric hindrance owing to the presence of the tetrasubstituted carbon, undergoes a spontaneous cleavage of the acyl group that affords finally tetrasubstituted α-aminophosphonate **18**.

Nevertheless, attempts to isolate compound **18** failed due to its decomposition during the workup, but the identity of thioamide **22** was confirmed by NMR of the crude reaction. ^31^P NMR showed the disappearance of the starting imine (δ_P_ = 6.6 ppm) and the formation of a major compound with a chemical shift at δ_P_ = 18.7 ppm. On the other hand, ^1^H NMR showed two clear doublets at δ_H_ = 3.90 ppm (^3^*J*_PH_ = 10.7 Hz) and δ_H_ = 3.79 ppm (^3^*J*_PH_ = 10.8 Hz), typical for the diastereotopic methoxy groups at the phosphonate, that suggest the formation of a stereogenic carbon close to the phosphorus atom and a broad triplet that interchanges with D_2_O, at δ_H_ = 8.63 ppm (^1^*J*_NH_ = 4.0 Hz), that may correspond to the NH of thioamide group, where the proton is coupled with the quadrupolar nucleus of ^14^N. More importantly, ^13^C NMR shows a doublet for the quaternary C-P (DEPT) at δ_C_ = 58.6 ppm (^1^*J*_PH_ = 167.2 Hz), and the characteristic chemical shift for the C=S group of thioamides at δ_C_ = 199.2 ppm. A similar result was obtained using thiobenzoic acid instead of thioacetic acid.

Additionally, the hydrolysis the phosphonate group to its phosphonic acid derivative **23** can be performed under mild conditions in chloroform by the treatment of **13b** with trimethylsilyl bromide at room temperature. The subsequent aqueous workup yields α-aminophosphonic acid **23** in almost quantitative yield (Scheme 6).

### 2.2. Biological Results

In vitro cytotoxicity of tetrasubstituted α-aminophosphonate derivatives **13**, **15** and **23** was evaluated by testing their antiproliferative activities against A549 cell line (carcinomic human alveolar basal epithelial cell). Cell counting kit (CCK-8) assay was used for the evaluation of growth inhibition. Moreover, nonmalignant MRC5 lung fibroblasts were tested for studying selective toxicity [34] and chemotherapeutic doxorubicin is used as reference value. The cell proliferation inhibitory activity is shown as IC_50_ values (Table 1).

In a preliminary study, we tested the cytotoxicity of simple phenyl substituted α-aminophosphonates **13a**–**j** as lead compounds. Although no grown inhibition activity was observed for dimethyl and diethylphosphonates **13a**,**f** (Table 1, Entries 1–2), dibenzylphosphonate **13g** showed some cytotoxicity against A549 cell line with an IC_50_ value of 16.46 ± 1.49 µM and, interestingly, very good selectivity was also obtained towards MRC5 nonmalignant cell line (Table 1, Entry 3). Besides, bulkier di-*iso*-propylphosphonate **13h**, derived from cyclohexyl isocyanide, presented an IC_50_ value of 19.72 ± 3.70 µM (Table 1, Entry 4).

Then we studied the introduction of substituents at the aromatic ring of tetrasubstituted aminophosphonates **13**. Scarce cytotoxic effect was found for *para*-nitrophenyl substituted substrate **13k**, bearing an electron poor aromatic group (Table 1, Entry 5). Although the effect of the introduction of fluorine atoms in the structure of organic compounds is rather difficult to predict, very often it leads to increased activities [35,36,37]. For this reason, next we tested the in vitro cytotoxicity of fluorine containing α-aminophosphonates **13n**. However, *ortho*-fluorophenyl and *para*-trifluoromethylphenyl substituted substrates **13n**,**r** presented IC_50_ values higher than 50 µM (Table 1, Entries 6, 8). Interestingly, thioether derived α-aminophosphonate **13p**, showed a considerable antiproliferative activity with an IC_50_ value of 14.56 ± 2.53 µM and a very good selectivity towards MRC5 cell line (Table 1, Entry 7). Phosphorated α-amido amide **15a** bearing a *para*-trifluomethylphenyl substituent showed better toxicity than its parent compound **13r** with an IC_50_ value of 28.76 ± 3.20 µM and a good selectivity towards nonmalignant cells (Table 1, Entry 8 vs. Entry 9). Finally phosphonic acid derivative **23** did not provide any toxicity against A549 cell line (Table 1, Entry 10).

## 3. Materials and Methods

### 3.1. Chemistry

#### 3.1.1. General Experimental Information

Solvents for extraction and chromatography were technical grade. All solvents used in reactions were freshly distilled from appropriate drying agents before use. All other reagents were recrystallized or distilled as necessary. All reactions were performed under an atmosphere of dry nitrogen. Analytical TLC was performed with silica gel 60 F_254_ plates. Visualization was accomplished by UV light. ^1^H, ^13^C, ^31^P and ^19^F-NMR spectra were recorded on a Varian Unity Plus (Varian Inc, NMR Systems, Palo Alto, CA, USA) (at 300 MHz, 75 MHz, 120 MHz and 282 MHz respectively) and on a Bruker Avance 400 (Bruker BioSpin GmbH, Rheinstetten, Germany) (at 400 MHz for ^1^H, and 100 MHz for ^13^C). Chemical shifts (δ) are reported in ppm relative to residual CHCl_3_ (δ = 7.26 ppm for ^1^H and δ = 77.16 ppm for ^13^C NMR) and using phosphoric acid (50%) as external reference (δ = 0.0 ppm) for ^31^P NMR spectra. Coupling constants (*J*) are reported in Hertz. Data for ^1^H NMR spectra are reported as follows: chemical shift, multiplicity, coupling constant, integration. Multiplicity abbreviations are as follows: s = singlet, d = doublet, t = triplet, q = quartet, m = multiplet). ^13^C NMR peak assignments were supported by distortionless enhanced polarization transfer (DEPT). High resolution mass spectra (HRMS) were obtained by positive-ion electrospray ionization (ESI). Data are reported in the form *m*/*z* (intensity relative to base = 100). Infrared spectra (IR) were taken in a Nicolet iS10 Thermo Scientific spectrometer (Thermo Scientific Inc., Waltham, Massachusetts, MA, USA) as neat solids. Peaks are reported in cm^−1^.

#### 3.1.2. Compounds Purity Analysis

All synthesized compounds were analyzed by HPLC to determine their purity. The analyses were performed on Agilent 1260 infinity HPLC system (Agilent, Santa Clara, CA, USA) using a CHIRALPAK® IA column (5μm, 0.54 cm ø × 25 cm, Daicel Chiral Technologies, Illkirch Cedex, France) at room temperature. All the tested compounds were dissolved in dichloromethane, and 5 μL of the sample was loaded onto the column. Ethanol and heptane were used as the mobile phase, and the flow rate was set at 1.0 mL/min. The maximal absorbance at the range of 190–400 nm was used as the detection wavelength. The purity of all the tested α-aminophosphonate derivatives **13**, **15** and α-aminophosphonaic acid **23** is >95%, which meets the purity requirement by the Journal.

#### 3.1.3. Experimental Procedures and Characterization Data for Compounds **13**, **15**, **16** and **23**

##### General Procedure for the Synthesis *N*-Tosyl α-Iminophosphonates **10**

Following a literature procedure, [28,31] to a solution of the corresponding tetrasubstituted *N*-tosyl α-aminophosphonate (10 mmol) in CH_2_Cl_2_ (30 mL) was added trichloroisocyanuric acid (6.97 g, 30 mmol). The resulting suspension was stirred at 0 °C until disappearance of the starting *N*-tosyl α-aminophosphonate, as monitored by ^31^P NMR (14 to 48 h). The solid residue was eliminated by filtration to afford a clear solution of intermediate *N*-chloro α-aminophosphonate and then, poly(4-vinylpyridine) (3.0 g), previously dried at 100 °C overnight, was added. The resulting suspension was stirred under reflux overnight and the reaction was then filtered and concentrated under reduced pressure. The resulting yellow oily crude was purified by crystallization from diethyl ether.

##### General Procedure for the Synthesis *N*-Trityl α-Iminophosphonate **14**

Following a literature procedure, [33] *N*-bromosuccinimide (178 mg, 1 mmol) was added on a solution of dimethyl ((tritylamino)methyl)phosphonate (457 mg, 1 mmol) in CCl_4_ (3 mL). The mixture was stirred in quartz flask under UV light until the disappearance of starting α-aminophosphonate as monitored by ^31^P-NMR (δ_H_ 30.9 to 10.1 ppm). The resulting suspension was filtered under inert atmosphere to afford a clear solution of dimethyl (*E*)-((tritylimino)methyl)phosphonate that can be used without any further workup.

##### General Procedure for the Ugi Reaction of α-Phosphorated Ketimines **10** and **14**

A mixture of α-iminophosphonate 10 or 14 (1 mmol), phenylacetic acid (**11**, 136 mg, 1 mmol) and isocyanide 12 (1.1 mmol) in ichloromethane (3 mL) was stirred at room temperature until disappearance of the starting iminophosphonate 10 as monitored by ^31^P-NMR. The reaction was concentrated under vacuum and the resulting crude residue was purified by crystallization (Dichomethane/Hexanes 1:3), yielding α-aminophosphonates **13**, **15** or **16**. In some cases, a purification by column chromatography was necessary as detailed for each compound.

*Dimethyl (2-(cyclohexylamino)-1-((4-methylphenyl)sulfonamido)-2-oxo-1-phenylethyl)phosphonate* (**13a**). The general procedure was followed, using dimethyl (*E*)-(phenyl(tosylimino)methyl) phosphonate (**10a**, 367 mg, 1 mmol), phenylacetic acid (**11**, 136 mg, 1 mmol) and cyclohexyl isocyanide (**12a**, 136 μL, 1.1 mmol) to afford 420 mg (85%) of **13a** after 1 h as white crystals after crystallization (Dichloromethane/Hexanes 1:3). M.p.: 206–208 °C. ^1^H-NMR (400 MHz, CDCl_3_) δ 7.25–7.15 (m, 3H, 3 × CH_Ar_), 7.16 (d, ^3^*J*_HH_ = 8.3 Hz, 2H, 2 × CH_Ar_), 7.05 (d, ^3^*J*_HH_ = 8.5 Hz, 2H, 2 × CH_Ar_), 7.00 (d, ^3^*J*_HH_ = 8.3 Hz, 2H, 2 × CH_Ar_), 6.76 (d, ^3^*J*_HH_ = 6.4 Hz, 1H, NHCO), 6.47 (d, ^3^*J*_PH_ = 8.2 Hz, 1H, NHTs), 3.99 (d, ^3^*J*_PH_ = 10.7 Hz, 3H, OCH_3_), 3.80 (d, ^3^*J*_PH_ = 10.5 Hz, 3H, OCH_3_), 3.77 (m, 1H, CHCy), 2.33 (s, 3H, CH_3_Ts), 1.91–1.44 (m, 4H, CH_2_Cy), 1.39–0.92 (m, 6H, 3 × CH_2_Cy) ppm. ^13^C-NMR {^1^H} (101 MHz, CDCl_3_) δ 166.1 (C=O), 142.5 (C_quat_Ts), 139.2 (d, ^4^*J*_PC_ = 1.6 Hz, C_quat_Ts), 131.9 (d, ^2^*J*_PC_ = 1.8 Hz, C_quat_ Ph), 130.2 (d, ^3^J_PC_ = 8.3 Hz, 2 × CH_Ar_ Ph), 129.1 (2 × CH_Ar_), 128.7 (CH_Ar_), 127.9 (2 × CH_Ar_), 126.5 (2 × CH_Ar_), 68.5 (d, ^1^*J*_PC_ = 157.2 Hz, C_quat_-P), 55.8 (d, ^2^*J*_PC_ = 8.2 Hz, OCH_3_), 55.2 (d, ^2^*J*_PC_ = 7.5 Hz, OCH_3_), 49.8 (CHCy), 32.3 (CH_2_Cy), 32.1 (CH_2_Cy), 25.4 (CH_2_Cy), 24.6 (CH_2_Cy), 24.5 (CH_2_Cy), 21.6 (CH_3_Ts) ppm. ^31^P-NMR (121 MHz, CDCl_3_) δ 20.5 ppm. FTIR (neat) ν_max_: ν = 3426 (N-H st) 3333 (N-H st) 1678 (C=O st) 1256 (P=O st) 1333 (S=O st sym) 1164 (S=O st as) cm^−1^. HRMS (ESI-TOF) *m*/*z*: [M + H]^+^ calcd for C_23_H_32_N_2_O_6_PS 495.1719, Found 495.1718.

*Methyl (2-(dimethoxyphosphoryl)-2-((4-methylphenyl)sulfonamido)-2-phenylacetyl)glycinate* (**13b**). The general procedure was applied starting from dimethyl (*E*)-(phenyl(tosylimino)methyl)phosphonate (**10a**, 367 mg, 1 mmol), phenylacetic acid (**11**, 136 mg, 1 mmol) and methyl 2-isocyanoacetate (**12b**, 100 μL, 1.1 mmol) to afford 378 mg (78%) of **13b** after 1 h as a white solid after a chromatography column (Hexanes/AcOEt 1:1), followed by crystallization (Dichloromethane/Hexanes 1:3). M.p.: 130–132 °C. ^1^H-NMR (400 MHz, CDCl_3_) δ 7.37–7.30 (m, 2H, 2 × CH_Ar_), 7.23–7.16 (m, 3H, 3 × CH_Ar_), 7.15 (d, ^3^*J*_HH_ = 5.5 Hz, 1H, NHCO), 7.11–6.95 (m, 4H, 4 × CH_Ar_), 6.64 (d, ^3^*J*_PH_ = 6.8 Hz, 1H, NHTs), 4.00 (d, ^3^*J*_HH_ = 5.5 Hz, 2H, CH_2_), 3.91 (d, ^3^*J*_PH_ = 10.8 Hz, 3H, POCH_3_), 3.78 (d, ^3^*J*_PH_ = 10.6 Hz, 3H, POCH_3_), 3.68 (s, 3H, COCH_3_), 2.33 (s, 3H, CH_3_Ts) ppm. ^13^C-NMR {^1^H} (101 MHz, CDCl_3_) δ 169.2 (C=O), 167.8 (C=O), 142.7 (C_quat_Ts), 139.0 (C_quat_Ts), 131.5 (C_quat_Ph), 130.2 (d, ^3^*J*_PC_ = 7.8 Hz, 2 × CH_Ar_Ph), 129.1 (2 × CH_Ar_), 128.9 (CH_Ar_), 128.0 (2 × CH_Ar_), 126.7 (2 × CH_Ar_), 69.0 (d, ^1^*J*_PC_ = 155.3 Hz, C_quat_-P), 55.6 (d, ^2^*J*_PC_ = 8.0 Hz, POCH_3_), 55.4 (d, ^2^*J*_PC_ = 7.5 Hz, POCH_3_), 52.5 (COCH_3_), 42.2 (CH_2_), 21.6 (CH_3_Ts) ppm. ^31^P-NMR (121 MHz, CDCl_3_) δ 18.9 ppm. FTIR (neat) ν_max_: ν = 3437 (N-H st) 3323 (N-H st) 1675 (C=O st) 1266 (P=O st) 1338 (S=O st sym) 1165 (S=O st as) cm^−1^. HRMS (ESI-TOF) *m*/*z*: [M+H]^+^ calcd for C_20_H_26_N_2_O_8_PS 485.1147, Found 485.1149.

*Dimethyl (2-(benzylamino)-1-((4-methylphenyl)sulfonamido)-2-oxo-1-phenylethyl)phosphonate* (**13c**). The general procedure was applied starting from dimethyl (*E*)-(phenyl(tosylimino)methyl) phosphonate (**10a**, 367 mg, 1 mmol), phenylacetic acid (**11**, 136 mg, 1 mmol) and (isocyanomethyl) benzene (**12c**, 134 μL, 1.1 mmol) to afford 402 mg (80%) of **13c** after 1 h as white crystals after a chromatography column (Hexanes/AcOEt 7:3), followed by crystallization (Dichloromethane/Hexanes 1:3). M.p.: 150–152 °C. ^1^H-NMR (400 MHz, CDCl_3_) δ 7.32–7.09 (m, 11H, 11 × CH_Ar_), 7.08–6.99 (m, 3H, 3 × CH_Ar_), 6.97 (d, ^3^*J*_HH_ = 6.3 Hz, 1H, NHCO), 6.74 (d, ^3^*J*_PH_ = 6.6 Hz, 1H, NHTs), 4.51 (dd, ^2^*J*_HH_ = 14.9 Hz, ^3^*J*_HH_ = 6.2 Hz, 1H, CH_A_CH_B_), 4.35 (dd, ^2^*J*_HH_ = 14.9 Hz, ^3^*J*_HH_ = 5.7 Hz, 1H, CH_A_CH_B_), 3.91 (d, ^3^*J*_PH_ = 10.8 Hz, 3H, OCH_3_), 3.67 (d, ^3^*J*_PH_ = 10.6 Hz, 3H, OCH_3_), 2.34 (s, 3H, CH_3_Ts) ppm. ^13^C-NMR {^1^H} (101 MHz, CDCl_3_) δ 167.2 (C=O), 142.7 (C_quat_Ts), 139.2 (C_quat_Ts), 137.2 (C_quat_Ph), 132.0 (C_quat_Ph), 130.1 (d, ^3^*J*_PC_ = 8.1 Hz, 2 × CH_Ar_Ph), 129.1 (2 × CH_Ar_), 128.8 (CH_Ar_), 128.7 (2 × CH_Ar_), 128.1 (2 × CH_Ar_), 127.8 (2 × CH_Ar_), 127.7 (CH_Ar_), 126.6 (2 × CH_Ar_), 68.8 (d, ^1^*J*_PC_ = 156.0 Hz, C_quat_-P), 55.6 (d, ^2^*J*_PC_ = 8.0 Hz, OCH_3_), 55.1 (d, ^2^*J*_PC_ = 7.6 Hz, OCH_3_), 44.7 (CH_2_), 21.6 (CH_3_Ts) ppm. ^31^P-NMR (121 MHz, CDCl_3_) δ 19.3 ppm. FTIR (neat) ν_max_: ν = 3428 (N-H st) 3341 (N-H st) 1675 (C=O st) 1255 (P=O st) 1332 (S=O st sym) 1160 (S=O st as) cm^−1^. HRMS (ESI-TOF) *m*/*z*: [M + H]^+^ calcd for C_24_H_28_N_2_O_6_PS 503.1406, Found 503.1411.

*Diethyl (2-(cyclohexylamino)-1-((4-methylphenyl)sulfonamido)-2-oxo-1-phenylethyl)phosphonate* (**13d**). The general procedure was applied starting from diethyl (*E*)-(phenyl(tosylimino)methyl) phosphonate (**10b**, 395 mg, 1 mmol), phenylacetic acid (**11**, 136 mg, 1 mmol) and cyclohexyl isocyanide (**12a**, 136 μL, 1.1 mmol) to afford 434 mg (83%) of **13d** after 6 h as white crystals after a chromatography column (Hexanes/AcOEt 8:2), followed by crystallization (Dichloromethane/Hexanes 1:3). M.p.: 122–124 °C. ^1^H NMR (400 MHz, CDCl_3_) δ 7.28–7.10 (m, 5H, 5 × CH_Ar_), 7.05–6.95 (m, 4H, 4 × CH_Ar_), 6.73 (d, ^3^*J*_HH_ = 6.4 Hz, 1H, NHCO), 6.54 (d, ^3^*J*_PH_ = 8.1 Hz, 1H, NHTs), 4.47–4.28 (m, 2H, OCH_2_), 4.27–4.06 (m, 2H, OCH_2_), 3.76 (m, 1H, CHCy), 2.33 (s, 3H, CH_3_Ts), 1.91–1.47 (m, 6H, 3 × CH_2_Cy), 1.40 (t, ^3^*J*_HH_ = 7.0 Hz, 3H, CH_3_CH_2_), 1.27 (t, ^3^*J*_HH_ = 7.0 Hz, 3H, CH_3_CH_2_) 1.30–0.94 (m, 4H, 2 × CH_2_Cy) ppm. ^13^C NMR {^1^H} (101 MHz, CDCl_3_) δ 166.4 (C=O), 142.4 (C_quat_Ts), 139.3 (C_quat_Ts), 132.1 (C_quat_Ph), 130.4 (d, ^3^*J*_PC_ = 7.8 Hz, 2 × CH_Ar_Ph), 129.0 (2 × CH_Ar_), 128.6 (CH_Ar_), 127.8 (2 × CH_Ar_), 126.6 (2 × CH_Ar_), 68.6 (d, ^1^*J*_PC_ = 156.2 Hz, C_quat_-P), 65.5 (d, ^2^*J*_PC_ = 8.3 Hz, OCH_2_), 65.0 (d, ^2^*J*_PC_ = 7.6 Hz, OCH_2_), 49.7 (CHCy), 32.2 (CH_2_Cy), 32.2 (CH_2_Cy), 25.4 (CH_2_Cy), 24.6 (CH_2_Cy), 24.5 (CH_2_Cy), 21.5 (CH_3_Ts), 16.6 (d, ^3^*J*_PC_ = 5.7 Hz, CH_3_CH_2_), 16.4 (d, ^3^*J*_PC_ = 5.8 Hz, CH_3_CH_2_) ppm. ^31^P-NMR (121 MHz, CDCl_3_) δ 17.1 ppm. FTIR (neat) ν_max_: ν = 3432 (N-H st) 3340 (N-H st) 1672 (C=O st) 1253 (P=O st) 1338 (S=O st sym) 1165 (S=O st as) cm^−1^. HRMS (ESI-TOF) *m*/*z*: [M + H]^+^ calcd for C_25_H_36_N_2_O_6_PS 523.2032, Found 523.2034.

*Methyl (2-(diethoxyphosphoryl)-2-((4-methylphenyl)sulfonamido)-2-phenylacetyl)glycinate* (**13e**). The general procedure was applied starting from diethyl (*E*)-(phenyl(tosylimino)methyl) phosphonate (**10b**, 395 mg, 1 mmol), phenylacetic acid (**11**, 136 mg, 1 mmol) and methyl 2-isocyanoacetate (**12b**, 100 μL, 1.1 mmol) to afford 453 mg (88%) of **13e** after 6 h as white crystals after crystallization (Dichloromethane/Hexanes 1:3). M.p.: 147–149 °C. ^1^H-NMR (400 MHz, CDCl_3_) δ 7.36–7.29 (m, 2H, 2 × CH_Ar_), 7.21–7.11 (m, 4H, 3 × CH_Ar_ + NHCO), 7.06–6.92 (m, 4H, 4 × CH_Ar_), 6.61 (d, ^3^*J*_PH_ = 6.9 Hz, 1H, NHTs), 4.34–4.19 (m, 2H, OCH_2_), 4.18–4.04 (m, 2H, OCH_2_), 3.96 (dd, ^3^*J*_HH_ = 5.5 Hz, ^5^*J*_PH_ = 2.7 Hz, 2H, NCH_2_), 3.65 (s, 3H, OCH_3_), 2.29 (s, 3H, CH_3_Ts), 1.30 (t, ^3^*J*_HH_ = 6.9 Hz, 3H, CH_3_CH_2_), 1.19 (t, ^3^*J*_HH_ = 7.0 Hz, 3H, CH_3_CH_2_) ppm. ^13^C-NMR {^1^H} (101 MHz, CDCl_3_) δ 169.2 (C=O), 167.9 (C=O), 142.6 (C_quat_Ts), 139.09 (d, ^4^*J*_PC_ = 1.4 Hz, C_quat_Ts), 131.6 (C_quat_Ph), 130.4 (d, ^3^*J*_PC_ = 7.6 Hz, 2 × CH_Ar_Ph), 129.0 (2 × CH_Ar_), 128.7 (CH_Ar_), 127.8 (2 × CH_Ar_), 126.7 (2 × CH_Ar_), 69.1 (d, ^1^*J*_PC_ = 153.9 Hz, C_quat_-P), 65.4 (d, ^2^*J*_PC_ = 8.0 Hz, OCH_2_), 65.1 (d, ^2^*J*_PC_ = 7.5 Hz, OCH_2_), 52.4 (OCH_3_), 42.2 (NCH_2_), 21.5 (CH_3_Ts), 16.5 (d, ^3^*J*_PC_ = 5.7 Hz, CH_3_CH_2_), 16.4 (d, ^3^*J*_PC_ = 5.6 Hz, CH_3_CH_2_) ppm. ^31^P-NMR (121 MHz, CDCl_3_) δ 16.7 ppm. FTIR (neat) ν_max_: ν = 3425 (N-H st) 3331 (N-H st) 1678 (C=O st) 1256 (P=O st) 1332 (S=O st sym) 1165 (S=O st as) cm^−1^. HRMS (ESI-TOF) *m*/*z*: [M + H]^+^ calcd for C_22_H_30_N_2_O_8_PS 513.1460, Found 513.1462.

*Diethyl (2-(benzylamino)-1-((4-methylphenyl)sulfonamido)-2-oxo-1-phenylethyl)phosphonate* (**13f**). The general procedure was applied starting from diethyl (*E*)-(phenyl(tosylimino)methyl) phosphonate (**10b**, 395 mg, 1 mmol), phenylacetic acid (**11**, 136 mg, 1 mmol) and (isocyanomethyl) benzene (**12c**, 134 μL, 1.1 mmol) to afford 411 mg (78%) of **13f** after 6 h as white crystals after crystallization (Dichloromethane/Hexanes 1:3). M.p.: 140–142 °C. ^1^H-NMR (400 MHz, CDCl_3_) δ 7.34–7.14 (m, 10H, 10 × CH_Ar_), 7.11–6.97 (m, 5H, 4 × CH_Ar_ + NHCO), 6.69 (d, ^3^*J*_PH_ = 5.7 Hz, 1H, NHTs), 4.52 (dd, ^2^*J*_HH_ = 14.9, ^3^*J*_HH_ = 6.2 Hz, 1H, NCH_A_CH_B_), 4.44–4.22 (m, 3H, NCH_A_CH_B_ + OCH_2_), 4.17–3.95 (m, 2H, OCH_2_), 2.35 (s, 3H, CH_3_Ts), 1.35 (t, ^3^*J*_HH_ = 7.0 Hz, 3H, CH_3_CH_2_), 1.17 (t, ^3^*J*_HH_ = 7.0 Hz, 3H, CH_3_CH_2_) ppm. ^13^C-NMR {^1^H} (101 MHz, CDCl_3_) δ 167.5 (C=O), 142.5 (C_quat_Ts), 139.3 (C_quat_Ts), 137.2 (C_quat_Ph), 132.1 (C_quat_Ph), 130.2 (d, ^3^*J*_PC_ = 7.7 Hz, 2 × CH_Ar_Ph), 129.1 (2 × CH_Ar_), 128.7 (2 × CH_Ar_), 128.6 (CH_Ar_), 128.0 (2 × CH_Ar_), 127.8 (2 × CH_Ar_), 127.7 (CH_Ar_), 126.6 (2 × CH_Ar_), 68.9 (d, ^1^*J*_PC_ = 155.0 Hz, C_quat_-P), 65.5 (d, ^2^*J*_PC_ = 8.2 Hz, OCH_2_), 65.0 (d, ^2^*J*_PC_ = 7.5 Hz, OCH_2_), 44.7 (NCH_2_), 21.6 (CH_3_Ts), 16.5 (d, ^3^*J*_PC_ = 5.7 Hz, CH_3_CH_2_), 16.3 d, ^3^*J*_PC_ = 5.7 Hz, CH_3_CH_2_) ppm. ^31^P-NMR (121 MHz, CDCl_3_) δ 16.9 ppm. FTIR (neat) ν_max_: ν = 3433 (N-H st) 3342 (N-H st) 1679 (C=O st) 1257 (P=O st) 1330 (S=O st sym) 1162 (S=O st as) cm^−1^. HRMS (ESI-TOF) *m*/*z*: [M + H]^+^ calcd for C_26_H_32_N_2_O_6_PS 531.1719, Found 531.1728.

*Dibenzyl (2-(benzylamino)-1-((4-methylphenyl)sulfonamido)-2-oxo-1-phenylethyl)phosphonate* (**13g**). The general procedure was applied starting from dibenzyl (*E*)-(phenyl(tosylimino)methyl) phosphonate (**10c**, 520 mg, 1 mmol), phenylacetic acid (**11**, 136 mg, 1 mmol) and cyclohexyl isocyanide (**12a**, 136 μL, 1.1 mmol) to afford 583 mg (90%) of **13g** after 6 h as white crystals after a chromatography column (Hexanes/AcOEt 8:2), followed by crystallization (Ether/Pentane 1:3). M.p.: 80–82 °C. ^1^H-NMR (400 MHz, CDCl_3_) δ 7.44–7.25 (m, 10H, 10 × CH_Ar_), 7.24–7.13 (m, 5H, 5 × CH_Ar_), 7.04 (t, ^3^*J*_HH_ = 7.7 Hz, 2H, 2 × CH_Ar_), 6.96 (d, ^3^*J*_HH_ = 8.1 Hz, 2H, 2 × CH_Ar_), 6.82 (d, ^3^*J*_HH_ = 6.6 Hz, 1H, NHCO), 6.44 (d, ^3^*J*_PH_ = 8.0 Hz, 1H, NHTs), 5.31–5.09 (m, 2H, OCH_2_), 5.06–4.91 (m, 2H, OCH_2_), 3.77–3.63 (m, 1H, CHCy), 2.33 (s, 3H, CH_3_Ts), 1.80–0.81 (m, 10H, 5 × CH_2_Cy) ppm. ^13^C-NMR {^1^H} (101 MHz, CDCl_3_) δ 166.1 (C=O), 142.5 (C_quat_Ts), 139.2 (C_quat_Ts), 135.9 (d, ^3^*J*_PC_ = 5.2 Hz, C_quat_Bn), 135.7 (d, ^3^*J*_PC_ = 6.0 Hz, C_quat_Bn), 131.9 (C_quat_Ph), 129.1 (2 × CH_Ar_), 128.8–128.5 (m, 9 × CH_Ar_), 128.3 (2 × CH_Ar_), 128.2 (2 × CH_Ar_), 127.9 (2 × CH_Ar_), 126.6 (2 × CH_Ar_), 70.6 (d, ^2^*J*_PC_ = 8.0 Hz, OCH_2_), 70.1 (d, ^2^*J*_PC_ = 7.5 Hz, OCH_2_), 68.9 (d, ^1^*J*_PC_ = 156.4 Hz, C_quat_-P), 49.8 (CHCy), 32.1 (CH_2_Cy), 32.0 (CH_2_Cy), 25.4 (CH_2_Cy), 24.5 (CH_2_Cy), 24.5 (CH_2_Cy), 21.6 (CH_3_Ts) ppm. ^31^P-NMR (121 MHz, CDCl3) δ (ppm): 17.7 ppm. FTIR (neat) ν_max_: ν = 3417 (N-H st) 3324 (N-H st) 1677 (C=O st) 1259 (P=O st) 1336 (S=O st sym) 1162 (S=O st as) cm^−1^. HRMS (ESI-TOF) *m*/*z*: [M + H]^+^ calcd for C_35_H_40_N_2_O_6_PS 647.2346, Found 647.2348.

*Di-iso-propyl (2-(cyclohexylamino)-1-((4-methylphenyl)sulfonamido)-2-oxo-1-phenylethyl)phosphonate* (**13h**). The general procedure was applied starting from di-*iso*-propyl (*E*)-(phenyl(tosylimino)methyl) phosphonate (**10d**, 423 mg, 1 mmol), phenylacetic acid (**11**, 136 mg, 1 mmol) and cyclohexyl isocyanide (**12a**, 136 μL, 1.1 mmol) to afford 474 mg (86%) of **13h** after 14 h as white crystals after a chromatography column (Hexanes/AcOEt 8:2), followed by crystallization (Dichloromethane/Hexanes 1:3). M.p.: 141–143 °C. ^1^H-NMR (400 MHz, CDCl_3_) δ 7.32–7.27 (m, 2H, 2 × CH_Ar_), 7.21–7.09 (m, 3H, 3 × CH_Ar_), 7.05–6.93 (m, 4H, 4 × CH_Ar_), 6.63 (d, ^3^*J*_HH_ = 7.3 Hz, 1H, NHCO), 6.58 (d, ^3^*J*_PH_ = 8.0 Hz, 1H, NHTs), 4.88 (m, 1H, CH_3_CH), 4.76 (m, 1H, CH_3_CH), 3.77 (m, 1H, CHCy), 2.32 (s, 3H, CH_3_Ts), 1.93–1.44 (m, 6H, 3 × CH_2_Cy), 1.37 (d, ^3^*J*_HH_ = 6.1 Hz, 3H, CH_3_CH), 1.36 (d, ^3^*J*_HH_ = 6.2 Hz, 3H, CH_3_CH), 1.24 (d, ^3^*J*_HH_ = 6.1 Hz, 3H, CH_3_CH), 1.15 (d, ^3^*J*_HH_ = 6.2 Hz, 3H, CH_3_CH), 1.40–0.97 (m, 4H, 2 × CH_2_Cy) ppm. ^13^C-NMR {^1^H} (101 MHz, CDCl_3_) δ 166.4 (C=O), 142.3 (C_quat_Ts), 139.4 (C_quat_Ts), 132.4 (C_quat_Ph), 130.3 (d, ^3^*J*_PC_ = 7.7 Hz, 2 × CH_Ar_), 129.0 (2 × CH_Ar_), 128.4 (CH_Ar_), 127.7 (2 × CH_Ar_), 126.7 (2 × CH_Ar_), 73.9 (d, ^2^*J*_PC_ = 7.8 Hz, CH_3_CH), 73.8 (d, ^2^*J*_PC_ = 8.2 Hz, CH_3_CH), 68.8 (d, ^1^*J*_PC_ = 156.0 Hz, C_quat_-P), 49.6 (CHCy), 32.2 (2 × CH_2_Cy), 25.5 (CH_2_Cy), 24.5 (d, ^3^*J*_PC_ = 7.2 Hz, CH_3_CH), 24.3 (d, ^3^*J*_PC_ = 6.5 Hz, CH_3_CH), 24.2 (2 × CH_2_Cy), 23.9 (d, ^3^*J*_Pc_ = 5.5 Hz, CH_3_CH), 23.5 (d, ^2^*J*_Pc_ = 5.8 Hz, CH_3_CH), 21.5 (CH_3_Ts) ppm. ^31^P-NMR (121 MHz, CDCl_3_) δ 15.8 ppm. FTIR (neat) ν_max_: ν = 3424 (N-H st) 3345 (N-H st) 1680 (C=O st) 1258 (P=O st) 1335 (S=O st sym) 1165 (S=O st as) cm^−1^. HRMS (ESI-TOF) *m*/*z*: [M + H]^+^ calcd for C_27_H_40_N_2_O_6_PS 551.2345, Found 551.2348.

*Methyl (2-(di-iso-propoxyphosphoryl)-2-((4-methylphenyl)sulfonamido)-2-phenylacetyl)glycinate* (**13i**). The general procedure was applied starting from diisopropyl (*E*)-(phenyl(tosylimino)methyl) phosphonate (**10d**, 423 mg, 1 mmol), phenylacetic acid (**11**, 136 mg, 1 mmol) and methyl 2-isocyanoacetate (**12b**, 100 μL, 1.1 mmol) to afford 335 mg (62%) of **13i** after 14 h as white crystals after a chromatography column (Hexanes/AcOEt 6:4), followed by crystallization (Dichloromethane/Hexanes 1:3). M.p.: 146–148 °C. ^1^H-NMR (400 MHz, CDCl_3_) δ 7.44–7.35 (m, 2H, 2 × CH_Ar_), 7.25–7.14 (m, 4H, 3 × CH_Ar_ + NHCO), 7.08–6.98 (m, 4H, 4 × CH_Ar_), 6.43 (d, ^3^*J*_PH_ = 7.8 Hz, 1H, NHTs), 4.82 (hept, ^3^*J*_HH_ = 6.0, 1H, CHCH_3_), 4.74 (hept, ^3^*J*_HH_ = 6.3 Hz, 1H, CHCH_3_), 4.03 (d, ^3^*J*_HH_ = 5.2 Hz, 2H, CH_2_), 3.72 (s, 3H, OCH_3_), 2.33 (s, 3H, CH_3_Ts), 1.34 (d, ^3^*J*_HH_ = 6.0 Hz, 3H, CH_3_CH), 1.31 (d, ^3^*J*_HH_ = 6.3 Hz, 3H, CH_3_CH), 1.23 (d, ^3^*J*_HH_ = 6.3 Hz, 3H, CH_3_CH), 1.11 (d, ^3^*J*_HH_ = 6.0 Hz, 3H, CH_3_CH) ppm. ^13^C-NMR {^1^H} (101 MHz, CDCl_3_) δ 169.3 (C=O), 167.9 (d, ^2^*J*_PC_ = 1.3 Hz, C=O), 142.6 (C_quat_Ts), 139.0 (d, ^4^*J*_PC_ = 1.3 Hz, C_quat_Ts), 131.8 (d, ^2^*J*_PC_ = 1.1 Hz, C_quat_Ph), 130.3 (d, ^3^*J*_PC_ = 7.5 Hz, 2 × CH_Ar_), 129.1 (2 × CH_Ar_), 128.6 (CH_Ar_), 127.7 (2 × CH_Ar_), 126.8 (2 × CH_Ar_), 74.4 (d, ^2^*J*_PC_ = 7.9 Hz, CH), 74.1 (d, ^2^*J*_PC_ = 8.0 Hz, CH), 69.5 (d, ^1^*J*_PC_ = 152.4 Hz, C_quat_-P), 52.5 (OCH_3_), 42.2 (CH_2_), 24.3 (d, ^3^*J*_PC_ = 5.0 Hz, CH_3_CH), 24.2 (d, ^3^*J*_PC_ = 4.5 Hz, CH_3_CH), 23.8 (d, ^3^*J*_PC_ = 5.7 Hz, CH_3_CH), 23.4 (d, ^3^*J*_PC_ = 6.0 Hz, CH_3_CH), 21.6 (CH_3_Ts) ppm. ^31^P-NMR (121 MHz, CDCl_3_) δ 15.1 ppm. FTIR (neat) ν_max_: ν = 3425 (N-H st) 3334 (N-H st) 1674 (C=O st) 1254 (P=O st) 1330 (S=O st sym) 1163 (S=O st as) cm^−1^. HRMS (ESI-TOF) *m*/*z*: [M + H]^+^ calcd for C_24_H_34_N_2_O_8_PS 541.1773, Found 541.1778.

*Di-iso-propyl (2-(benzylamino)-1-((4-methylphenyl)sulfonamido)-2-oxo-1-phenylethyl)phosphonate* (**13j**). The general procedure was applied starting from di-*iso*-propyl (*E*)-(phenyl(tosylimino)methyl) phosphonate (**10d**, 423 mg, 1 mmol), phenylacetic acid (**11**, 136 mg, 1 mmol) and (isocyanomethyl) benzene (**12c**, 134 μL, 1.1 mmol) to afford 464 mg (83%) of **13j** after 14 h as white crystals after a chromatography column (Hexanes/AcOEt 8:2), followed by crystallization (Dichloromethane/Hexanes 1:3). M.p.: 100–102 °C. ^1^H-NMR (400 MHz, CDCl_3_) δ 7.39–7.31 (m, 2H, 2 × CH_Ar_), 7.30–7.13 (m, 8H, 7 × CH_Ar_ + NHCO), 7.06–6.98 (m, 5H, 5 × CH_Ar_), 6.59 (d, ^3^*J*_PH_ = 7.4 Hz, 1H, NHTs), 4.85 (hept, ^3^*J*_HH_ = 6.2 Hz, 1H, CH), 4.64 (hept, ^3^*J*_HH_ = 6.1 Hz, 1H, CH), 4.55 (dd, ^2^*J*_HH_ = 14.9, ^3^*J*_HH_ = 6.3 Hz, 1H, CH_A_CH_B_), 4.33 (dd, ^2^*J*_HH_ = 14.9, ^3^*J*_HH_ = 5.3 Hz, 1H, CH_A_CH_B_), 2.33 (s, 3H, CH_3_Ts), 1.33 (d, ^3^*J*_HH_ = 6.1 Hz, 3H, CH_3_CH), 1.30 (d, ^3^*J*_HH_ = 6.2 Hz, 3H, CH_3_CH), 1.14 (d, ^3^*J*_HH_ = 6.1 Hz, 3H, CH_3_CH), 1.06 (d, ^3^*J*_HH_ = 6.2 Hz, 3H, CH_3_CH) ppm. ^13^C-NMR {^1^H} (101 MHz, CDCl_3_) δ 167.5 (C=O), 142.5 (C_quat_Ts), 139.3 (d, ^4^*J*_PC_ = 1.3 Hz, C_quat_Ts), 137.2 (C_quat_Ph), 132.4 (C_quat_Ph), 130.2 (d, ^3^*J*_PC_ = 7.6 Hz, 2 × CH_Ar_), 129.0 (2 × CH_Ar_), 128.7 (2 × CH_Ar_), 128.5 (CH_Ar_), 127.8 (2 × CH_Ar_), 127.8 (2 × CH_Ar_), 127.7 (CH_Ar_), 126.8 (2 × CH_Ar_), 74.1 (d, ^2^*J*_PC_ = 8.0 Hz, 2 × CH), 69.1 (d, ^1^*J*_PC_ = 154.6 Hz, C_quat_-P), 44.6 (CH_2_), 24.3 (d, ^2^*J*_PC_ = 3.6 Hz, CH_3_CH), 24.1 (d, ^2^*J*_PC_ = 3.0 Hz, CH_3_CH), 23.8 (d, ^2^*J*_Pc_ = 5.8 Hz, CH_3_CH), 23.5 (d, ^2^*J*_PC_ = 5.7 Hz, CH_3_CH), 21.6 (CH_3_Ts) ppm. ^31^P-NMR (121 MHz, CDCl_3_) δ 15.4 ppm. FTIR (neat) ν_max_: ν = 3423 (N-H st) 3336 (N-H st) 1678 (C=O st) 1256 (P=O st) 1331 (S=O st sym) 1162 (S=O st as) cm^−1^.HRMS (ESI-TOF) *m*/*z*: [M + H]^+^ calcd for C_28_H_36_N_2_O_6_PS 559.2032, Found 559.2038.

*Dimethyl (2-(cyclohexylamino)-1-((4-methylphenyl)sulfonamido)-1-(4-nitrophenyl)-2-oxoethyl)phosphonate* (**13k**). The general procedure was applied starting from dimethyl (*E*)-((4-nitrophenyl)(tosylimino)methyl)phosphonate (**10e**, 412 mg, 1 mmol), phenylacetic acid (**11**, 136 mg, 1 mmol) and cyclohexyl isocyanide (**12a**, 136 μL, 1mmol) to afford 418 mg (78%) of **13k** after 1 h as white crystals after a chromatography column (Hexanes/AcOEt 7:3), followed by crystallization (Dichloromethane/Hexanes 1:3). M.p.: 100–102 °C. ^1^H-NMR (400 MHz, CDCl_3_) δ 7.88–7.84 (m, 2H, 2 × CH_Ar_), 7.52 (d, ^3^*J*_HH_ = 8.7 Hz, 2H, 2 × CH_Ar_), 7.22 (d, ^3^*J*_HH_ = 8.7 Hz, 2H, 2 × CH_Ar_), 7.08–7.01 (m, 2H, 2 × CH_Ar_), 6.79 (d, ^3^*J*_HH_ = 8.2 Hz, 1H, NHCO), 6.77 (d, ^3^*J*_PH_ = 8.1 Hz, 1H, NHTos), 3.96 (d, ^3^*J*_PH_ = 10.9 Hz, 3H, OCH_3_), 3.76 (d, ^3^*J*_PH_ = 10.7 Hz, 3H, OCH_3_), 3.76 (m, 1H, CHCy), 2.37 (s, 3H, CH_3_Ts), 1.93–1.51 (m, 4H, 2 × CH_2_Cy), 1.41–0.97 (m, 6H, 3 × CH_2_Cy) ppm. ^13^C-NMR {^1^H} (101 MHz, CDCl_3_) δ 165.0 (C=O), 147.5 (C_quat-_NO_2_), 143.6 (C_quat_ Ts), 139.7 (C_quat_ Ts), 138.8 (C_quat_Ar), 130.9 (d, ^3^*J*_PC_ = 7.6 Hz, 2 × CH_Ar_), 129.3 (2 × CH_Ar_), 126.5 (2 × CH_Ar_), 122.8 (2 × CH_Ar_), 67.8 (d, ^1^*J*_PC_ = 153.9 Hz, C_quat_-P), 56.0 (d, ^2^*J*_PC_ = 8.0 Hz, OCH_3_), 55.6 (d, ^2^*J*_PC_ = 7.6 Hz, OCH_3_), 50.0 (CHCy), 32.3 (CH_2_Cy), 32.2 (CH_2_Cy), 25.4 (CH_2_Cy), 24.6 (CH_2_Cy), 24.5 (CH_2_Cy), 21.6 (CH_3_Ts) ppm. ^31^P-NMR (121 MHz, CDCl3) δ 18.8 ppm. FTIR (neat) ν_max_: ν = 3422 (N-H st) 3346 (N-H st) 1677 (C=O st) 1263 (P=O st) 1348 (S=O st sym) 1165 (S=O st as) cm^−1^. HRMS (ESI-TOF) *m*/*z*: [M + H]^+^ calcd for C_23_H_31_N_3_O_8_PS 540.1569, Found 540.1571.

*Dimethyl (1-(4-bromophenyl)-2-(cyclohexylamino)-1-((4-methylphenyl)sulfonamido)-2-oxoethyl)phosphonate* (**13l**). The general procedure was applied starting from dimethyl (*E*)-((4-bromophenyl)(tosylimino)methyl)phosphonate (**10f**, 446 mg, 1 mmol), phenylacetic acid (**11**, 136 mg, 1 mmol) and cyclohexyl isocyanide (**12a**, 136 μL, 1 mmol) to afford 495 mg (87%) of **13l** after 1 h as white crystals after crystallization (Dichloromethane/Hexanes 1:3). M.p.: 150–152 °C. ^1^H-NMR (400 MHz, CDCl_3_) δ 7.16 (d, ^3^*J*_HH_ = 8.3 Hz, 2 × CH_Ar_), 7.09 (m, seen as s, 4H, 4 × CH_Ar_), 7.04 (d, ^3^*J*_HH_ = 8.3 Hz, 2H, 2 × CH_Ar_), 6.77 (d, ^3^*J*_HH_ = 7.4 Hz, 1H, NHCO), 6.60 (d, ^3^*J*_PH_ = 8.2 Hz, 1H, NHTs), 3.96 (d, ^3^*J*_PH_ = 10.8 Hz, 3H, OCH_3_), 3.77 (d, ^3^*J*_PH_ = 10.6 Hz, 3H, OCH_3_), 3.74 (m, 1H, CHCy), 2.36 (s, 3H, CH_3_Ts), 1.91–1.48 (m, 4H, 2 × CH_2_Cy), 1.41–0.95 (m, 6H, 3 × CH_2_Cy) ppm. ^13^C-NMR {^1^H} (101 MHz, CDCl_3_) δ 165.7 (C=O), 143.0 (C_quat_Ts), 138.9 (C_quat_Ts), 131.8 (d, ^3^*J*_PC_ = 8.1 Hz, 2 × CH_Ar_), 131.1 (C_quat_Ar), 130.9 (2 × CH_Ar_), 129.2 (2 × CH_Ar_), 126.5 (2 × CH_Ar_), 123.4 (C_quat-_Br), 67.9 (d, ^1^*J*_PC_ = 155.9 Hz, C_quat_-P), 55.9 (d, ^2^*J*_PC_ = 8.2 Hz, OCH_3_), 55.4 (d, ^2^*J*_PC_ = 7.5 Hz, OCH_3_), 49.8 (CHCy), 32.3 (CH_2_Cy), 32.2 (CH_2_Cy), 25.4 (CH_2_Cy), 24.6 (CH_2_Cy), 24.5 (CH_2_Cy), 21.6 (CH_3_Ts) ppm. ^31^P-NMR (121 MHz, CDCl3) δ 19.2 ppm. FTIR (neat) ν_max_: ν = 3421 (N-H st) 3320 (N-H st) 1670 (C=O st) 1250 (P=O st) 1343 (S=O st sym) 1165 (S=O st as) cm^−1^. HRMS (ESI-TOF) *m*/*z*: [M + H]^+^ calcd for C_23_H_31_BrN_2_O_6_PS 573.0824, Found 573.0830.

*Dimethyl (1-(4-chlorophenyl)-2-(cyclohexylamino)-1-((4-methylphenyl)sulfonamido)-2-oxoethyl)phosphonate* (**13m**). The general procedure was applied starting from dimethyl (*E*)-((4-chlorophenyl)(tosylimino)methyl) phosphonate (**10g**, 402 mg, 1 mmol), phenylacetic acid (**11**, 136 mg, 1 mmol) and cyclohexyl isocyanide (**12a**, 136 μL, 1 mmol) to afford 422 mg (80%) of **13m** after 1 h as white crystals after a chromatography column (Hexanes/AcOEt 7:3), followed by crystallization (Dichloromethane/Hexanes 1:3). M.p.: 165–167 °C. ^1^H-NMR (400 MHz, CDCl_3_) δ 7.19–7.12 (m, 4H, 4 × CH_Ar_), 7.04 (d, ^3^*J*_HH_ = 8.2 Hz, 2H, 2 × CH_Ar_), 6.95 (d, ^3^*J*_HH_ = 8.2 Hz, 2H, 2 × CH_Ar_), 6.76 (d, ^3^*J*_HH_ = 7.2 Hz, 1H, NHCO), 6.58 (d, ^3^*J*_PH_ = 8.1 Hz, 1H, NHTs), 3.97 (d, ^3^*J*_PH_ = 10.8 Hz, 3H, OCH_3_), 3.78 (d, ^3^*J*_PH_ = 10.6 Hz, 3H, OCH_3_), 3.75 (m, 1H, CHCy), 2.36 (s, 3H, CH_3_Ts), 1.91–0.95 (m, 10H, 5 × CH_2_Cy) ppm. ^13^C-NMR {^1^H} (101 MHz, CDCl_3_) δ 165.7 (C=O), 142.9 (C_quat_Ts), 139.0 (C_quat_Ts), 135.0 (C_quat_Ar), 131.5 (d, ^3^*J*_PC_ = 7.7 Hz, 2 × CH_Ar_), 130,6 (C_quat-_Cl), 129.1 (2 × CH_Ar_), 128.0 (2 × CH_Ar_), 126.5 (2 × CH_Ar_), 67.8 (d, ^1^*J*_PC_ = 155.9 Hz, C_quat_-P), 55.9 (d, ^2^*J*_PC_ = 8.2 Hz, OCH_3_), 55.4 (d, ^2^*J*_PC_ = 7.5 Hz, OCH_3_), 49.8 (CHCy), 32.3 (CH_2_Cy), 32.2 (CH_2_Cy), 25.4 (CH_2_Cy), 24.6 (CH_2_Cy), 24.5 (CH_2_Cy), 21.6 (CH_3_Ts) ppm. ^31^P-NMR (121 MHz, CDCl3) δ (ppm): 19.2 ppm. FTIR (neat) ν_max_: ν = 3417 (N-H st) 3319 (N-H st) 1670 (C=O st) 1258 (P=O st) 1353 (S=O st sym) 1160 (S=O st as) cm^−1^. HRMS (ESI-TOF) *m*/*z*: [M + H]^+^ calcd for C_23_H_31_ClN_2_O_6_PS 529.1331, Found 529.1335.

*Dimethyl (2-(cyclohexylamino)-1-(2-fluorophenyl)-1-((4-methylphenyl)sulfonamido)-2-oxoethyl)phosphonate* (**13n**). The general procedure was applied starting from dimethyl (*E*)-((2-fluorophenyl)(tosylimino)methyl) phosphonate (**10h**, 385 mg, 1 mmol), phenylacetic acid (**11**, 136 mg, 1 mmol) and cyclohexyl isocyanide (**12a**, 136 μL, 1 mmol) to afford 370 mg (73%) of **13n** after 1 h as white crystals after crystallization (Dichloromethane/Hexanes 1:3). M.p.: 203–205 °C. ^1^H-NMR (400 MHz, CDCl_3_) δ 7.88 (t, ^3^*J*_FH_ = 7.3 Hz, 1H, CH_Ar_), 7.22 (m, 1H, CH_Ar_), 7.18–7.09 (m, 3H, 3 × CH_Ar_), 7.03–6.93 (m, 3H, 3 × CH_Ar_), 6.41 (m, 1H, NHCO), 5.98 (d, ^3^*J*_PH_ = 8.2 Hz, 1H, NHTs), 4.03 (d, ^3^*J*_PH_ = 10.7 Hz, 3H, OCH_3_), 3.87 (d, ^3^*J*_PH_ = 10.5 Hz, 3H, OCH_3_), 3.75 (m, 1H, CHCy), 2.32 (s, 3H, CH_3_Ts), 1.88–1.44 (m, 4H, 2 × CH_2_Cy), 1.38–0.78 (m, 6H, 3 × CH_2_Cy) ppm. ^13^C-NMR {^1^H} (101 MHz, CDCl_3_) δ 165.3 (C=O), 160.6 (dd, ^1^*J*_FC_ = 252.7, ^3^*J*_PC_ = 13.1 Hz, CF). 142.7 (C_quat_Ts), 138.4 (d, C_quat_Ts), 134.2 (d, ^3^*J*_FC_ = 5.2 Hz, CH_Ar_), 131.5 (d, ^3^*J*_FC_ = 9.1 Hz, CH_Ar_), 129.0 (2 × CH_Ar_), 126.6 (2 × CH_Ar_), 123.9 (d, ^4^*J*_FC_ = 3.4 Hz, CH_Ar_), 119.9 (dd, ^2^*J*_FC_ = 10.8, ^2^*J*_PC_ = 4.6 Hz, C_quat_Ar), 115.7 (d, ^2^*J*_FC_ = 22.5 Hz, CH_Ar_), 64.6 (d, ^1^*J*_PC_ = 158.9 Hz, C_quat_-P), 56.2 (d, ^2^*J*_PC_ = 8.5 Hz, OCH_3_), 55.6 (d, ^2^*J*_PC_ = 7.7 Hz, OCH_3_), 50.0 (CHCy), 32.2 (2 × CH_2_Cy), 25.4 (CH_2_Cy), 24.7 (CH_2_Cy), 24.6 (CH_2_Cy), 21.6 (CH_3_Ts) ppm. ^31^P-NMR (121 MHz, CDCl3) δ 17.8 ppm. ^19^F-NMR (282 MHz, CDCl_3_) δ −105.5 ppm. FTIR (neat) ν_max_: ν = 3417 (N-H st) 3319 (N-H st) 1672 (C=O st) 1268 (P=O st) 1352 (S=O st sym) 1165 (S=O st as) cm^−1^. HRMS (ESI-TOF) *m*/*z*: [M + H]^+^ calcd for C_23_H_31_FN_2_O_6_PS 513.1624, Found 513.1630.

*Dimethyl (2-(cyclohexylamino)-1-((4-methylphenyl)sulfonamido)-2-oxo-1-(perfluorophenyl)ethyl)phosphonate* (**13o**). The general procedure was applied starting from dimethyl (*E*)-((perfluorophenyl)(tosylimino)methyl) phosphonate (**10i**, 457 mg, 1 mmol), phenylacetic acid (**11**, 136 mg, 1 mmol) and cyclohexyl isocyanide (**12a**, 136 μL, 1 mmol) to afford 514 mg (88%) of **13o** after 1 h as white crystals after crystallization (Dichloromethane/Hexanes 1:3). M.p. (Dichloromethane/Hexanes) = 189–191 °C. ^1^H-NMR (400 MHz, CDCl_3_) δ 7.31 (d, ^3^*J*_HH_ = 8.1 Hz, 2H, 2 × CH_Ar_), 7.11 (d, ^3^*J*_HH_ = 8.1 Hz, 2H, 2 × CH_Ar_), 6.92 (d, ^3^*J*_HH_ = 7.3 Hz, 1H, NHCO), 6.40 (br s, 1H, NHTs), 4.01 (d, ^3^*J*_PH_ = 10.9 Hz, 3H, OCH_3_), 3.82 (d, ^3^*J*_PH_ = 10.7 Hz, 3H, OCH_3_), 3.79–3.69 (m, 1H, CHCy), 2.36 (s, 3H, CH_3_Ts), 1.96–0.94 (m, 10H, 5 × CH_2_Cy) ppm. ^13^C-NMR {^1^H} (101 MHz, CDCl_3_) δ 164.0 (C=O), 146.4 (m, 2 × CF), 144.0 (C_quat_Ts), 142.1 (m, 2 × CF), 137.8 (C_quat_Ts), 137.5 (m, CF), 129.2 (2 × CH_Ar_), 126.3 (2 × CH_Ar_), 108.9 (m, C_quat_Ar), 61.2 (d, ^1^*J*_PC_ = 159.7 Hz, C_quat_-P), 56.4 (d, ^2^*J*_PC_ = 8.6 Hz, OCH_3_), 55.9 (d, ^2^*J*_PC_ = 8.1 Hz, OCH_3_), 50.3 (CHCy), 32.3 (CH_2_Cy), 31.9 (CH_2_Cy), 25.4 (CH_2_Cy), 24.7 (CH_2_Cy), 24.6 (CH_2_Cy), 21.5 (CH_3_Ts) ppm. ^31^P-NMR (121 MHz, CDCl3) δ 15.9 ppm. ^19^F-NMR (282 MHz, CDCl_3_) δ −129.4, −151.7, −162.3 ppm. FTIR (neat) ν_max_: ν = 3417 (N-H st) 3316 (N-H st) 1677 (C=O st) 1268 (P=O st) 1330 (S=O st sym) 1165 (S=O st as) cm^−1^. HRMS (ESI-TOF) *m*/*z*: [M + H]^+^ calcd for C_23_H_27_F_5_N_2_O_6_PS 585.1249, Found 584.1166.

*Dimethyl (2-(cyclohexylamino)-1-((4-methylphenyl)sulfonamido)-2-oxo-1-(4-((trichloromethyl)thio)phenyl)ethyl)phosphonate* (**13p**). The general procedure was applied starting from dimethyl (*E*)-((tosylimino)(4-((trichloromethyl)thio)phenyl)methyl) phosphonate (**10j**), phenylacetic acid (**11**, 136 mg, 1 mmol) and cyclohexyl isocyanide (**12a**, 136 μL, 1 mmol) to afford 580 mg (92%) of **13p** after 14 h as white crystals after a chromatography column (Hexanes/AcOEt 7:3), followed by crystallization (Dichloromethane/Hexanes 1:3). M.p.: 148–150 °C. ^1^H-NMR (400 MHz, CDCl_3_) δ 7.49 (d, ^3^*J*_HH_ = 8.4 Hz, 2H, 2 × CH_Ar_), 7.42 (d, ^3^*J*_HH_ = 8.4 Hz, 2H, 2 × CH_Ar_), 7.24 (d, ^3^*J*_HH_ = 8.1 Hz, 2H, 2 × CH_Ar_), 7.06 (d, ^3^*J*_HH_ = 8.1 Hz, 2H, 2 × CH_Ar_), 6.75 (d, ^3^*J*_HH_ = 6.8 Hz, 1H, NHCO), 6.58 (d, ^3^*J*_PH_ = 8.2 Hz, 1H, NHTs), 3.96 (d, ^3^*J*_PH_ = 10.8 Hz, 3H, OCH_3_), 3.77 (d, ^3^*J*_PH_ = 10.6 Hz, 3H, OCH_3_), 3.71 (m, 1H, CHCy), 2.33 (s, 3H, CH_3_Ts), 1.91–1.45 (m, 4H, 2 × CH_2_Cy), 1.39–0.94 (m, 6H, 3 × CH_2_Cy) ppm. ^13^C-NMR {^1^H} (101 MHz, CDCl_3_) δ 165.4 (C=O), 143.1 (C_quat_Ts), 139.1 (C_quat_Ts), 136.4 (2 × CH_Ar_), 136.2 (C_quat_Ar), 131.2 (C_quat_Ar) 130.8 (d, ^3^*J*_PC_ = 7.9 Hz, 2 × CH_Ar_), 129.3 (2 × CH_Ar_), 126.5 (2 × CH_Ar_), 98.2 (CCl_3_), 68.0 (d, ^1^*J*_PC_ = 155.9 Hz, C_quat_-P), 55.9 (d, ^2^*J*_PC_ = 8.1 Hz, OCH_3_), 55.4 (d, ^2^*J*_PC_ = 7.5 Hz, OCH_3_), 49.8 (CHCy), 32.2 (CH_2_Cy), 32.0 (CH_2_Cy), 25.4 (CH_2_Cy), 24.5 (CH_2_Cy), 24.4 (CH_2_Cy), 21.6 (CH_3_Ts) ppm. ^31^P-NMR (121 MHz, CDCl3) δ (ppm) 19.0 ppm. FTIR (neat) ν_max_: ν = 3412 (N-H st) 3337 (N-H st) 1677 (C=O st) 1265 (P=O st) 1353 (S=O st sym) 1160 (S=O st as) cm^−1^. HRMS (ESI-TOF) *m*/*z*: [M + H]^+^ calcd for C_24_H_31_Cl_3_N_2_O_6_PS_2_ 643.0427, Found 643.0430.

*Dimethyl (1-(3-chloro-4-methoxyphenyl)-2-(cyclohexylamino)-1-((4-methylphenyl)-sulfonamido)-2-oxoethyl)-phosphonate* (**13q**). The general procedure was followed, using dimethyl (*E*)-((3-chloro-4-methoxyphenyl)(tosylimino)-methyl) phosphonate (**10k**, 431 mg, 1 mmol), phenylacetic acid (**11**, 136 mg, 1 mmol) and cyclohexyl isocyanide (**12a**, 136 μL, 1 mmol) to afford 418 mg (75%) of **13q** after 14 h as white crystals after crystallization (Dichloromethane/Hexanes 1:3). M.p.: 115–117 °C. ^1^H-NMR (400 MHz, CDCl_3_) δ 7.45 (m, 1H, CH_Ar_), 7.34 (d, ^3^*J*_HH_ = 4.4 Hz, 1H, NHCO), 7.15 (d, ^3^*J*_HH_ = 8.0 Hz, 2H, 2 × CH_Ar_), 7.02 (d, ^3^*J*_HH_ = 8.0 Hz, 2H, 2 × CH_Ar_), 6.78–6.29 (m, 2H, 2 × CH_Ar_), 6.53 (d, ^3^*J*_PH_ = 7.5 Hz, 1H, NHTs), 4.04 (d, ^3^*J*_PH_ = 10.7 Hz, 3H, OCH_3_), 3.87 (s, 3H, OCH_3_), 3.83 (d, ^3^*J*_PH_ = 10.4 Hz, 3H, OCH_3_), 3.79 (m, 1H, CHCy), 2.34 (s, 3H, CH_3_Ts), 1.87–1.02 (m, 10H, 5 × CH_2_Cy) ppm. ^13^C-NMR {^1^H} (101 MHz, CDCl_3_) δ 165.9 (C=O), 155.2 (C_quat_), 143.2 (C_quat_ Ts), 138.7 (C_quat_ Ts), 132.0 (d, ^3^*J*_PC_ = 10.9 Hz, CH_Ar_), 131.0 (d, ^3^*J*_PC_ = 6.7 Hz, CH_Ar_), 129.2 (2 × CH_Ar_ Ts), 126.4 (2 × CH_Ar_ Ts), 124.3 (C_quat_), 121.9 (C_quat_), 110.2 (CH_Ar_), 67.5 (d, ^1^*J*_PC_ = 157.8 Hz, C_quat_-P), 56.3 (OCH_3_), 56.2 (d, ^2^*J*_PC_ = 7.4 Hz, OCH_3_), 55.4 (d, ^2^*J*_PC_ = 7.5 Hz, OCH_3_), 49.8 (CHCy), 32.3 (CH_2_Cy), 32.2 (CH_2_Cy), 25.4 (CH_2_Cy), 24.6 (CH_2_Cy), 24.5 (CH_2_Cy), 21.6 (CH_3_ Ts) ppm. ^31^P-NMR (121 MHz, CDCl_3_) δ 19.1 ppm. FTIR (neat) ν_max_: ν = 3434 (N-H st) 3323 (N-H st) 1677 (C=O st) 1258 (P=O st) 1334 (S=O st sym) 1160 (S=O st as) cm^−1^. HRMS (ESI-TOF) *m*/*z*: [M + H]^+^ calcd for C_24_H_33_ClN_2_O_7_PS 559.1436, Found 559.1437.

*Dimethyl (2-(cyclohexylamino)-2-oxo-1-(2-phenyl-N-tosylacetamido)-1-(4-(trifluoromethyl)phenyl)ethyl)phosphonate* (**15a**). The general procedure was applied starting from dimethyl (*E*)-((tosylimino)(4-(trifluoromethyl)phenyl)methyl) phosphonate (**10l**, 435 mg, 1 mmol), phenylacetic acid (**11**, 136 mg, 1 mmol) and cyclohexyl isocyanide (**12a**, 136 μL, 1 mmol) to afford 579 mg (85%) of **15a** after 1 h as white crystals after a chromatography column (Hexanes/AcOEt 7:3), followed by crystallization (Dichloromethane/Hexanes 1:3). M.p.: 84–86 °C. ^1^H-NMR (400 MHz, CDCl_3_) δ 8.20 (d, ^3^*J*_HH_ = 8.0 Hz, 2H, 2 × CH_Ar_), 7.78 (d, ^3^*J*_HH_ = 8.3 Hz, 2H, 2 × CH_Ar_), 7.57 (d, ^3^*J*_HH_ = 8.3 Hz, 2H, 2 × CH_Ar_), 7.36 (d, ^3^*J*_HH_ = 8.0 Hz, 2H, 2 × CH_Ar_), 7.28–7.17 (m, 5H, 5 × CH_Ar_), 5.95 (br s, 1H, NHCO), 4.20 (d, ^2^*J*_HH_ = 17.1 Hz, 1H, CH_A_CH_B_), 3.98 (d, ^2^*J*_HH_ = 17.1 Hz, 1H, CH_A_CH_B_), 3.84 (d, ^3^*J*_PH_ = 11.2 Hz, 3H, OCH_3_), 3.80–3.68 (m, 1H, CHCy), 3.50 (d, ^3^*J*_PH_ = 11.5 Hz, 3H, OCH_3_), 2.44 (s, 3H, CH_3_Ts), 1.98–0.90 (m, 10H, 6 × CH_2_Cy) ppm. ^13^C-NMR {^1^H} (101 MHz, CDCl_3_) δ 176.6 (C=O), 164.7 (d, ^2^*J*_PC_ = 5.2 Hz, C=O), 144.8 (C_quat_Ts), 138.5 C_quat_), 138.4 (C_quat_Ts), 133.9 (C_quat_CF_3_), 133.6 (C_quat_Ar), 130.2 (2 × CH_Ar_), 129.9 (2 × CH_Ar_), 129.2 (d, ^3^*J*_PC_ = 5.3 Hz, 2 × CH_Ar_), 128.4 (2 × CH_Ar_), 128.0 (2 × CH_Ar_), 127.3 (d, ^4^*J*_FC_ = 3.8 Hz, 2 × CH_Ar_), 125.1 (CH_Ar_), 123.9 (q, ^1^*J*_FC_ = 272.3 Hz, CF_3_), 79.4 (d, ^1^*J*_PC_ = 147.5 Hz, C_quat_-P), 55.4 (d, ^2^*J*_PC_ = 7.3 Hz, OCH_3_), 55.1 (d, ^2^*J*_PC_ = 7.7 Hz, OCH_3_), 49.2 (CHCy), 45.7 (CH_2_Bn), 32.4 (CH_2_Cy), 31.9 (CH_2_Cy), 25.5 (CH_2_Cy), 24.6 (CH_2_Cy), 24.4 (CH_2_Cy), 21.8 (CH_3_Ts) ppm. ^31^P-NMR (121 MHz, CDCl3) δ (ppm): 22.1 ppm. ^19^F-NMR (282 MHz, CDCl_3_) δ −63.1.ppm. FTIR (neat) ν_max_: ν = 3430 (N-H st) 1676 (C=O st) 1248 (P=O st) 1330 (S=O st sym) 1160 (S=O st as) cm^−1^.HRMS (ESI-TOF) *m*/*z*: [M + H]^+^ calcd for C_32_H_36_F_3_N_2_O_7_PS 680.1933, Found 680.1934.

*Dimethyl (2-(cyclohexylamino)-1-((4-methylphenyl)sulfonamido)-2-oxo-1-(4-(trifluoromethyl)phenyl)ethyl)phosphonate* (**13r**). Exposure of **15a** under air moisture for 48 h yields **13r** in quantitative yield as white crystals after crystallization (Dichloromethane/Hexanes 1:3). M.p.: 88–90 °C. ^1^H-NMR (400 MHz, CDCl_3_) δ 7.36 (d, ^3^*J*_HH_ = 7.8 Hz, 2H, 2 × CH_Ar_), 7.22 (d, ^3^*J*_HH_ = 7.8 Hz, 2H, 2 × CH_Ar_), 7.11 (d, ^3^*J*_HH_ = 8.4 Hz, 2H, 2 × CH_Ar_), 6.98 (d, ^3^*J*_HH_ = 8.4 Hz, 2H, 2 × CH_Ar_), 6.84 (d, ^3^*J*_HH_ = 7.8 Hz, 1H, NHCO), 6.70 (d, ^3^*J*_PH_ = 8.1 Hz, 1H, NHTs), 3.99 (d, ^3^*J*_PH_ = 10.8 Hz, 3H, OCH_3_), 3.80 (m, 1H, CHCy), 3.78 (d, ^3^*J*_PH_ = 10.6 Hz, 3H, OCH_3_), 2.32 (s, 3H, CH_3_Tos), 1.91–0.94 (m, 10H, 5 × CH_2_Cy) ppm. ^13^C-NMR {^1^H} (101 MHz, CDCl_3_) δ 165.4 (C=O), 143.0 (C_quat_Ts), 138.8 (C_quat_Ts), 135.8 (C_quat_), 130.6 (q seen as d, ^3^*J*_FC_ = 7.9 Hz, 2 × CH_Ar_), 130.7 (d, ^2^*J*_PC_ = 32.8 Hz, C_quat_CF_3_), 129.2 (2 × CH_Ar_), 126.3 (2 × CH_Ar_), 124.6 (d, ^4^*J*_FC_ = 3.7 Hz, 2 × CH_Ar_), 123.7 (q, ^1^*J*_FC_ = 272.5 Hz, CF_3_), 68.0 (d, ^1^*J*_PC_ = 155.2 Hz, C_quat_-P), 55.8 (d, ^2^*J*_PC_ = 8.1 Hz, OCH_3_), 55.5 (d, ^2^*J*_PC_ = 7.6 Hz, OCH_3_), 49.9 (CHCy), 32.2 (CH_2_Cy), 32.1 (CH_2_Cy), 25.3 (CH_2_Cy), 24.6 (CH_2_Cy), 24.6 (CH_2_Cy), 21.4 (CH_3_Tos) ppm. ^31^P-NMR (121 MHz, CDCl3) δ (ppm): 20.0 ppm. ^19^F-NMR (282 MHz, CDCl_3_) δ −63.5.ppm. FTIR (neat) ν_max_: ν = 3430 (N-H st) 3332 (N-H st) 1677 (C=O st) 1259 (P=O st) 1326 (S=O st sym) 1165 (S=O st as) cm^−1^.HRMS (ESI-TOF) *m*/*z*: [M + H]^+^ calcd for C_24_H_30_F_3_N_2_O_6_PS 562.1514, Found 562.1519.

*Dimethyl (2-(cyclohexylamino)-1-(4-fluorophenyl)-2-oxo-1-(2-phenyl-N-tosylacetamido)ethyl)phosphonate* (**15b**). The general procedure was followed, using dimethyl (*E*)-(4-fluorophenyl phenyl(tosylimino)methyl) phosphonate (**10m**, 385 mg, 1 mmol), phenylacetic acid (**11**, 136 mg, 1 mmol) and cyclohexyl isocyanide (**12a**, 136 μL, 1 mmol) to afford 536 mg (85%) of **15b** after 14 h as white crystals after column chromatography (Hexanes/AcOEt 8:2), followed by crystallization (Dichloromethane/Hexanes 1:3). M.p.: 90–92 °C. ^1^H-NMR (400 MHz, CDCl_3_) δ ^1^H-NMR (400 MHz, CDCl_3_) δ 8.26 (d, ^3^*J*_HH_ = 7.9 Hz, 2H, 2 × CH_Ar_), 7.05 (m, 2H, 2 × CH_Ar_), 7.37 (d, ^3^*J*_HH_ = 8.1 Hz, 2H, 2 × CH_Ar_), 7.28–7.18 (m, 5H, 5 × CH_Ar_), 7.03 (dd, seen as t, ^3^*J*_HH_ = 8.5 Hz, ^3^*J*_HH_ = 8.5 Hz, 2H, 2 × CH_Ar_), 5.93 (broad s, 1H, NH), 4.17 (d, ^2^*J*_HH_ = 17.1 Hz, 1H, CH_2_), 3.92 (d, ^2^*J*_HH_ = 17.1 Hz, 1H, CH_2_), 3.82 (d, ^3^*J*_PH_ = 11.2 Hz, 3H, OCH_3_), 3.73 (m, 1H, CHCy), 3.50 (d, ^3^*J*_PH_ = 11.5 Hz, 3H, OCH_3_), 2.44 (s, 3H, CH_3_Ts), 1.85–0.94 (m, 10H, 5 × CH_2_Cy) ppm. ^13^C-NMR {^1^H} (101 MHz, CDCl_3_) δ 176.4 (C=O), 165.3 (d, ^2^*J*_PC_ = 4.8 Hz, C=O), 162.4 (d, ^1^*J*_FC_ = 248.9 Hz, C_Ar_-F), 144.6 (C_quat_), 138.6 (C_quat_), 133.8 (C_quat_), 131.9 (m, C_quat_ + 2 × CH_Ar_), 130.2 (2 × CH_Ar_), 129.8 (2 × CH_Ar_), 128.4 (2 × CH_Ar_), 128.0 (2 × CH_Ar_), 127.2 (CH_Ar_), 115.2 (d, ^2^*J*_FC_ = 21.6 Hz, 2 × CH_Ar_), 78.9 (d, ^1^*J*_PC_ = 148.5 Hz, C_quat_-P), 55.2 (d, ^3^*J*_Pc_ = 7.2 Hz, CH_3_O), 54.9 (d, ^3^*J*_PC_ = 7.7 Hz, CH_3_O), 48.9 (CH Cy), 45.7 (CH_2_ Bn), 32.4 (CH_2_ Cy), 31.9 (CH_2_ Cy), 25.6 (CH_2_ Cy), 24.5 (CH_2_ Cy), 24.4 (CH_2_ Cy), 21.8 (CH_3_ Ts) ppm. ^31^P-NMR (121 MHz, CDCl3) δ 21.8 ppm. ^19^F NMR (282 MHz, CDCl_3_): δ −113.9 ppm. FTIR (neat) ν_max_: ν = 3426 (N-H st) 1677 (C=O st) 1263 (P=O st) 1348 (S=O st sym) 1156 (S=O st as) cm^−1^. HRMS (ESI-TOF) *m*/*z*: [M + H]^+^ calcd for C_31_H_36_FN_2_O_7_PS 631.2043, Found 631.2040.

*Dimethyl (2-(cyclohexylamino)-2-oxo-1-(2-phenylacetamido)ethyl)phosphonate* (**16**). The general procedure was followed, using dimethyl (*E*)-dimethyl (*E*)-(phenyl(tritylimino)methyl)phosphonate (**14**, 455 mg, 1 mmol), phenylacetic acid (**11**, 136 mg, 1 mmol) and cyclohexyl isocyanide (**12a**, 136 μL, 1 mmol) to afford 536 mg (65%) of **16** after 14 h as white crystals after column chromatography (Hexanes/AcOEt 8:2), followed by crystallization (Dichloromethane/Hexanes 1:3). M.p.: 121–122 °C. ^1^H-NMR (300 MHz, CDCl_3_) δ ^1^H-NMR (400 MHz, CDCl_3_) δ 7.43–7.20 (m, 5H, 5 × CH_Ar_), 5.25 (d, ^2^*J*_PH_ = 20.6 Hz, 1H, CH-P), 4.91 (br s, 2H, 2 × NH), 3.73 (br s, 3H, OCH_3_), 3.70 (br s, 3H, OCH_3_), 3.63 (m, 1H, CHCy), 3.58 (s, 2H, CH_2_), 1.84–1.65 (m, 4H, 2 × CH_2_Cy), 1.36–1.13 (m, 6H, 3 × CH_2_Cy) ppm. ^13^C-NMR {^1^H} (101 MHz, CD_3_OD) δ 174.3 (C=O), 172.0 (d, ^2^*J*_PC_ = 5.3 Hz, C=O), 135.3 (C_quat_), 129.0 (2 × CH_Ar_), 128.3 (2 × CH_Ar_), 126.7 (CH_Ar_), 53.5 (d, ^3^*J*_Pc_ = 6.8 Hz, CH_3_O), 53.3 (d, ^3^*J*_Pc_ = 6.4 Hz, CH_3_O), 50.5 (d, ^1^*J*_PC_ = 148.0 Hz, C_quat_-P), 49.0 (CH Cy), 42.0 (CH_2_ Bn), 32.2 (CH_2_ Cy), 32.1 (CH_2_ Cy), 25.3 (CH_2_ Cy), 24.7 (CH_2_ Cy), 24.6 (CH_2_ Cy) ppm. ^31^P-NMR (121 MHz, CDCl3) δ 21.0 ppm. FTIR (neat) ν_max_: ν = 3436 (N-H st) 3334 (N-H st) 1674 (C=O st) 1265 (P=O st) 1342 (S=O st sym) 1160 (S=O st as) cm^−1^.HRMS (ESI-TOF) *m*/*z*: [M + H]^+^ calcd for C_18_H_27_N_2_O_5_P, 383.1736, Found 383.1741.

##### Procedure for the Hydrolysis of Phosphonate Ester **13b**

A solution of α-aminophosphonate **13b**, (242 mg, 0.5 mmol) and bromotrimethylsilane (765 mg, 2.5 mmol) was stirred at room temperature in chloroform (3 mL) for 24 h until disappearance of the starting materials as monitored by ^31^P-NMR. Water (3 mL) was then added and the volatiles were distilled off at reduced pressure. The crude residue was crystallized from a mixture Dichloromethane/methanol (95:5) to afford 267 mg (99%) of **23** as a white solid. M.p.: 204–206 °C. ^1^H-NMR (400 MHz, D_2_O) δ 7.39 (d, ^3^*J*_HH_ = 7.3 Hz, 2H, 2 × CH_Ar_), 7.25 (d, ^3^*J*_HH_ = 8.5 Hz, 2H, 2 × CH_Ar_), 7.20–7.14 (m, 2H, 3H, 3 × CH_Ar_), 7.12–7.05 (m, 2H, 2 × CH_Ar_), 3.98 (d, ^2^*J*_HH_ = 17.9 Hz, 1H, CH_A_CH_B_), 3.72 (d, ^2^*J*_HH_ = 17.9 Hz, 1H, CH_A_CH_B_), 3.60 (s, 3H, OCH_3_), 2.20 (s, 3H, CH_3_Ts) ppm. ^13^C-NMR {^1^H} (101 MHz, D_2_O) δ 171.7 (C=O), 170.7 (C=O), 133.1 (d, ^2^*J*_PC_ = 4.3 Hz, C_quat_), 130.2 (d, ^3^*J*_PC_ = 5.8 Hz, 2 × CH_Ar_), 129.4 (2 × CH_Ar_), 128.1 (CH_Ar_), 127.4 (2 × CH_Ar_), 126.3 (2 × CH_Ar_), 69.9 (d, ^1^*J*_PC_ = 128.5 Hz, C_quat_-P), 52.8 (OCH_3_), 42.0 (CH_2_), 20.6 (CH_3_Ts) ppm. ^31^P-NMR (121 MHz, D_2_O) δ 13.1 ppm. FTIR (neat) ν_max_: ν = 3447 (N-H st) 3367 (N-H st) 1674 (C=O st) 1290 (P=O st) 1357 (S=O st sym) 1172 (S=O st as) cm^−1^. HRMS (ESI-TOF) *m*/*z*: [M + H]^+^ for C_18_H_21_N_2_O_8_PS 457.0834, Found 457.0831.

### 3.2. Biology

#### 3.2.1. Materials

Reagents and solvents were used as purchased without further purification. All stock solutions of the investigated compounds were prepared by dissolving the powered materials in appropriate amounts of Dimethylsulfoxide (DMSO). The final concentration of DMSO never exceeded 5% (*v*/*v*) in reactions. The stock solution was stored at 5 °C until it was used.

#### 3.2.2. Cell Culture

Human epithelial lung carcinoma cells (A549) (ATCC^®^ CCL-185™, ATCC - Manassas, VA, USA) were grown in Kaighn’s Modification of Ham’s F-12 Medium (ATCC^®^ 30-2004™, ATCC - Manassas, VA, United States) and lung fibroblast cells (MRC5) (ATCC^®^ CCL-171™, ATCC - Manassas, VA, USA) were grown in Eagle’s Minimum Essential Medium (EMEM, ATCC^®^ 30-2003™, ATCC - Manassas, VA, USA). Both were supplemented with 10% of fetal bovine serum (FBS) (Sigma-Aldrich, Spain) and with 1% of NORMOCIN solution (Thermo Fisher, Waltham, Massachusetts (MA), United States). Cells were incubated at 37 °C and 5% CO_2_ atmosphere, and were split every 3–4 days to maintain monolayer coverage. For cytotoxicity experiments, A549 cells were seeded in 96-well plates at a density of 2.5–3 × 10^3^ cells per well and incubated overnight to achieve 70% of confluence at the time of exposition to the cytotoxic compound.

#### 3.2.3. Cytotoxicity Assays

Cells were exposed to different concentrations of the cytotoxic compounds and were incubated for 48 h. Then, 10 µL of cell counting kit-8 was added into each well for additional two hours incubation at 37 °C. The absorbance of each well was determined by an Automatic Elisa Reader System (Thermo Scientific Multiskan FC Automatic Elisa Reader System, Thermo Scientific, Shangai, China) at 450 nm wavelength.

## 4. Conclusions

In conclusion, we report an efficient Ugi methodology using ketimines for the preparation of tetrasubstituted α-aminophosphonates holding a variety of substituents. Despite the difficulty often observed for the utilization of ketones or ketimines in Ugi reactions, α-phosphorated ketimines react under mild conditions to give the Ugi adducts after the spontaneous cleavage of the amide moiety. Clear evidences of the Ugi mechanism are provided, using thioacids. Moreover, obtained α-aminophosphonate derivatives **13g**, **13h**, **13p** and **15a** showed in vitro cytotoxicity inhibiting the growth of human tumor cell line A549 (carcinomic human alveolar basal epithelial cell), and a high selectivity toward MRC5 nonmalignant lung fibroblasts. As far as we know this is the first example of much hindered tetrasubstituted α-aminophosphonates showing antiproliferative activity.

## Data Availability

The data presented in this study are available in the Appendix A or on request from the corresponding author (^1^H, ^13^C, ^19^F and ^31^P-NMR and HRMS spectra and cytotoxicity essays).

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
