# Peer review of "Ugi Reaction on α-Phosphorated Ketimines for the Synthesis of Tetrasubstituted α-Aminophosphonates and Their Applications as Antiproliferative Agents"

_molecules, 2021, doi:10.3390/molecules26061654_

Round 1
Reviewer 1 Report
The Ugi reaction has been one of the most productive, abundantly studied and frequently used of all the multicomponent reactions. The present study discloses the use of phosporated N-tosylketimines under Ugi reaction conditions. The Ugi products are unstable, and the final compounds are actually the result of a two-component reaction - a carboxylic acid fragment is not incorporated in the final molecule. Still, the Ugi pathway was verified by authors. The present research is a competent study and will be interesting to a wide readership.
Author Response
We thank the very positive comments from referee Nr. 1. Regarding the revision of English language and style, all the authors have made a careful reading of the paper and some corrections have been made in order to improve the quality of the article
Reviewer 2 Report
This paper reports an interesting new application of the Ugi reaction on α-phosphorated N-tosyl ketimines, which react under mild conditions with different isocyanides, in the presence of carbocyclic acids, to form tetrasubstituted α-aminophosphonates. The reaction proceeds through a Ugi three component reaction but, due to the steric hindrance, the expected acyl amines undergo a spontaneous elimination of the acyl group.
Proof of mechanism is given with the use of a thioacids and extensive discussion of the mechanism and 1H NMR, 13C NMR and IR of the main product is furnished.
Evaluation of the compounds as potential anticancer agents is also furnished, and some new compounds showing selective activity towards malignant cell lines are also furnished.
The paper is pleasant to read. I recommend publication with minor revision as indicated below:
- At pag 2, authors mention the most straightforward method for the preparation of α-aminophosphonic acids (why they say “both” compounds?). [19,20] Here, insertion of an additional figure would help in comparing the proposed new method with existing methodologies.
- The use of CH2Cl2 is not desirable in a green chemistry point of view. Other solvents have been tested? Can a halide free solvent work as well? A comment on this issue would be desirable.
- In the Experimental Section number of compounds should be in bold character (please check).
- Supplementary: compound 13b contains some impurities (visible in the spectra). The signals at 162 ppm, 39 ppm and 30 ppm in the 13C NMR spectrum do not belong to the product. The same for 13c, 13n, 13r. If possible, further purification and replacement with more clean spectra would improve the overall quality of the manuscript.
- The Abstract is repeated twice.
Author Response
We thank the very positive comments from referee Nr. 2.
- At pag 2, authors mention the most straightforward method for the preparation of α-aminophosphonic acids (why they say “both” compounds?). [19,20] Here, insertion of an additional figure would help in comparing the proposed new method with existing methodologies.
By “both compounds”, we meant α-aminoacids and α-aminophosphonic acids, we agree that the sentence is not clear and we have added a clarification in this paragraph. We find unnecessary the addition of a scheme with the typical methods for the synthesis of α-aminophosphonates, since it is very repetitive in most of the papers on the topic of α-aminophosphonates. However if still such scheme is considered necessary, the classical chart showing the possible pathways for the synthesis of α-aminophosphonates through C-P, C-C and C-N bond formation could be added.
- The use of CH2Cl2 is not desirable in a green chemistry point of view. Other solvents have been tested? Can a halide free solvent work as well? A comment on this issue would be desirable.
We have added the following sentence “Due to the insolubility of the starting materials, using of other environmentally friendly solvents led to the formation of substrate 13a in lower yields and longer reaction times”
- In the Experimental Section number of compounds should be in bold character (please check)..
All the numbers of our compounds are now formatted in bold.
- Supplementary: compound 13b contains some impurities (visible in the spectra). The signals at 162 ppm, 39 ppm and 30 ppm in the 13C NMR spectrum do not belong to the product. The same for 13c, 13n, 13r. If possible, further purification and replacement with more clean spectra would improve the overall quality of the manuscript.
The impurities in 13C NMR come from the starting materials used for the preparation of the imine substrates. The spectra have been re-recorded after recrystallization.
- The Abstract is repeated twice.
True, and corrected.
Reviewer 3 Report
In this manuscript, the authors have investigated the variant of the Ugi reaction using α-phosphorated ketimines to synthesize hindered α-Aminophosphonates. All the synthesized compounds are well characterized. The synthetic approach is interesting and the manuscript could be considered after the following revision.
Line 33, Diversity-oriented synthesis is NOT a new model – the sentence should be modified
Scheme 1, write full name – Mumm rearrangement
It looks like Fig 1 is for peptide (amide) bond formation reaction whereas the text for Fig 1 and caption mentioned peptide cleavage. So either reaction arrows or text should be corrected
How did the authors come up with the initial reaction condition for 13a? Was it adopted from the literature or any optimization study was performed?
Line 198, replace ‘third party’ with an appropriate word like ‘third component/reactant’
The rationale behind the biological testing in the A549 cell line is not clear to me. How the cell line was selected and what is take away from the biological testing results?
Line 303, replace ‘crowded’ with ‘hindered’
Line 328, What was the instrument for the purification method? Ethanol and heptane used for column chromatography?
Line 331, Where is lactam derivatives in this manuscript? “The purity of all the tested lactam derivatives 4-12 and 16-21 is >95%, which…..”. Might be a mistake when copied from the previous papers.
Even though it is reported, at least show the general procedures for the synthesis of N-Tosyl and N-trityl α-ketiminophosphonates.
Line 342, remove ‘previous’
Important: Since the general procedure state that “….disappearance of the starting iminophosphonate 10 as monitored by 31P-NMR” – it is very essential to write the reaction time for each example.
All compound numbers should be BOLD in the experimental section as well.
Why carboxylic acid reactant is missing in the procedure for each example?
Why (Dichloromethane/Hexanes) is mentioned with M.p.?
Author Response
We thank the very positive comments from referee Nr. 3.
- Line 33, Diversity-oriented synthesis is NOT a new model – the sentence should be modified.
True. Once it WAS a new model, but not now. We have modified the sentence, accordingly.
- Scheme 1, write full name – Mumm rearrangement
Corrected, and also in Scheme 5.
- It looks like Fig 1 is for peptide (amide) bond formation reaction whereas the text for Fig 1 and caption mentioned peptide cleavage. So either reaction arrows or text should be corrected
Figure 1 has been modified accordingly.
- How did the authors come up with the initial reaction condition for 13a? Was it adopted from the literature or any optimization study was performed?
We used the typical reaction conditions for Ugi reaction but, in our case, due to the insolubility of the starting materials, CH2Cl2 was used as solvent. Some sentences have been added to clarify this point.
- Line 198, replace ‘third party’ with an appropriate word like ‘third component/reactant’
“Third party” has been modified by “third reactant”
- The rationale behind the biological testing in the A549 cell line is not clear to me. How the cell line was selected and what is take away from the biological testing results?
As routine work, we check the antiproliferative activity of our compounds on HEK293 (human embryonic kidney), MCF7 (human breast adenocarcinoma), HTB81 (human prostate carcinoma), HeLa (human epitheloid cervix carcinoma), RKO (human colon epithelial carcinoma), SKOV3 (human ovarian carcinoma) and A549 (carcinomic human alveolar basal epithelial cell) cell lines. In this case, we found only a relevant activity in A549 cell line which is what we report.
- Line 303, replace ‘crowded’ with ‘hindered’
“Crowded” has been replaced by the more correct term “hindered”
- Line 328, What was the instrument for the purification method? Ethanol and heptane used for column chromatography?
Ethanol/Heptane mixtures were used only for the determination of the purity of the samples by analytical HPLC. The purification of the compounds was performed by preparative column chromatography using Hexanes/AcOEt in rates that are described in the experimental section, followed by crystallizationthat.
- Line 331, Where is lactam derivatives in this manuscript? “The purity of all the tested lactam derivatives 4-12 and 16-21 is >95%, which…..”. Might be a mistake when copied from the previous papers.
True. A copy paste matter, followed by an oversight. Corrected
- Even though it is reported, at least show the general procedures for the synthesis of N-Tosyl and N-trityl α-ketiminophosphonates.
The general experimental procedures have been added.
- Line 342, remove ‘previous’
Removed.
- Important: Since the general procedure state that “….disappearance of the starting iminophosphonate 10 as monitored by 31P-NMR” – it is very essential to write the reaction time for each example.
Reaction times have been added for each compound in the experimental
- All compound numbers should be BOLD in the experimental section as well.
All compounds have been formatted in bold in the experimental section
- Why carboxylic acid reactant is missing in the procedure for each example?
The amount of phenylacetic acid used has been added for each example.
- Why (Dichloromethane/Hexanes) is mentioned with M.p.?
Although crystallization conditions are usually indicated for melting points, this information is already provided in the experimental procedure. Therefore, in order to avoid duplicated information we have removed this info from the M.p.